# Transferring Human Daily Activity Skills to Surgical Robots via Deep Successor Features

## Abstract

We propose a framework for surgical robot task learning that leverages human Activities of Daily Living (ADL) datasets to mitigate the scarcity of surgical training data. Surgical robot learning is uniquely constrained: datasets are limited, costly, and unsafe to collect at scale. In contrast, the robotics community has curated extensive ADL datasets capturing motor behaviors such as food preparation and tool use. Our key insight is that these datasets encode transferable visuomotor primitives—such as instrument manipulation and hand–eye coordination—that parallel the basic skills underlying surgical maneuvers. Inspired by how surgeons develop expertise by first mastering everyday skills before refining them in the operating room, we leverage ADL data to pretrain representations for surgical robot learning. To address task variability and embodiment differences, we design a modular deep successor feature architecture that learns predictive state representations from ADL tool-use and adapts them to surgical domains. Unlike prior approaches that depend solely on limited surgical data, our framework enables offline pretraining on abundant non-surgical datasets while supporting efficient reinforcement learning during deployment. We validate the framework on the da Vinci Research Kit (dVRK) in both simulation and real-world settings, showing that pretraining on ADLs accelerates adaptation with limited surgical data and improves sample efficiency compared to imitation learning and reinforcement learning baselines. While our current evaluation emphasizes a subset of fundamental surgical tasks, our results provide initial evidence that ADL pretraining offers a principled and scalable pathway toward data-efficient and safe autonomous surgical robot learning.

## 1 Introduction

Autonomous robotic surgery holds promise for advancing surgical practice by improving precision, dexterity, and consistency while supporting minimally invasive procedures that shorten recovery and reduce complications (Brodie & Vasdev, 2018). Automating skills such as suturing (Feng et al., 2023), tissue retraction (Singh et al., 2023), tumor resection (Ge et al., 2023), debridement (Sefati et al., 2021), ultrasound scanning (Deng et al., 2021), and suctioning (Richter et al., 2021) can reduce surgeon workload, standardize techniques, and enhance outcomes (Sridhar et al., 2017).

Although online reinforcement learning (RL) has been successfully applied to robotic manipulation tasks in general (Mahler et al., 2019; Zhu et al., 2020), the limited availability of surgical robots and the stringent safety requirements of clinical settings make direct RL training for surgical tasks particularly challenging (Xu et al., 2021; Barnoy et al., 2021). Consequently, most existing approaches rely on learning from demonstrations (LfD) (Kim et al., 2020; Pore et al., 2021; Amirshirzad et al., 2021). However, curated surgical datasets remain scarce and expensive, as they require surgeons to provide demonstrations, high-fidelity annotations, and specialized data collection pipelines that comply with strict safety regulations (Schmidgall et al., 2024; Kim et al., 2024; Sridhar et al., 2017). This mismatch between the high data demands of current methods and the practical constraints of clinical practice underscores a fundamental bottleneck for advancing surgical robot learning.

In contrast, human Activities of Daily Living (ADLs) datasets provide a rich source of fine-motor tool-use and object manipulation skills (Huang & Sun, 2019; Petrich et al., 2022). Although ADL and surgical tasks differ in context, they share structural similarities in motor dynamics. Prior work has shown that coordination in daily activities can support dexterous robotic skill development (Agarwal et al., 2021; Grollman & Billard, 2011), reflecting how surgeons build expertise on basic motor skills acquired through everyday practice (Kim et al., 2020). Surgical expertise is not innate but develops gradually: years of practicing basic motor behaviors lay the groundwork for dexterity, and continued engagement in surgical procedures further sharpens this proficiency (as in Fig. 1). Motivated by this parallel, we formulate surgical robot learning from ADL data as a problem of learning transferable representations that encode motor primitives. Instead of collecting costly surgical datasets, our approach leverages abundant ADL data to pretrain manipulation primitives and then adapts them to surgical domains with minimal supervision.

The prior work demonstrated that a single most relevant human daily activity skill could be transferred to a surgical task (Hu et al., 2025), showing the feasibility of cross-domain motor skill reuse. However, this single-skill transfer paradigm does not fully reflect natural human skill acquisition, where tasks are typically solved by recombining and adapting multiple prior experiences rather than relying on one closest match. Many manipulation behaviors share overlapping motion primitives and interaction patterns, and effective task execution often emerges from the coordinated composition of several foundational skills. Motivated by this observation, we move beyond single-skill reuse and propose a modular transfer framework that enables dynamic multi-skill composition for robotic surgical task learning.

In this work, we introduce a more general transfer mechanism based on modular successor features. Specifically, instead of identifying or manually selecting a single relevant ADL skill. We introduce a modular successor feature framework for surgical robot task learning through cross-domain motor transfer from human ADL datasets. Our method learns predictive and transferable state–action representations using deep successor features (Barreto et al., 2017; 2018), which decouple dynamics from rewards to enable motion reuse across tasks. By training on abundant ADL data, we extract motor primitives that can be adapted to surgical domains. During deployment, these representations support rapid policy refinement on real surgical robots, reducing reliance on scarce, task-specific demonstrations. This enables fast, safe, and data-efficient policy improvement, shifting the emphasis from task-specific supervision toward generalizable sensorimotor skill abstractions.

We evaluate the proposed framework on the da Vinci surgical robot, focusing on two clinically essential manipulators: the Patient Side Manipulator (PSM) for tool control and the Endoscopic Camera Manipulator (ECM) for camera guidance. While large-scale ADL datasets provide diverse tool-use and motor activity recordings, fine-grained hand motion trajectories are less frequently captured. To align with the available data, we select representative surgical tasks—PSM reaching a target manipulation area and ECM localizing/tracking a surgical view—that emphasize transferable motor dynamics. We conduct both simulation and real-world experiments, demonstrating that ADL-based pretraining substantially accelerates surgical robot task learning. To further analyze robustness, we perform ablation studies across different ADL categories, modular decomposition strategies, and module counts. Experimental results consistently show rapid and data-efficient adaptation, underscoring the strong potential of cross-domain motor transfer for scalable and safe surgical robot task learning. Our key contributions are:

(1) We present the first framework that systematically transfers motor skills from human ADL datasets to surgical robot task learning, alleviating the bottleneck of scarce expert surgical demonstrations and establishing ADLs as a scalable pretraining resource.

(2) We introduce a modular successor feature–based representation learning method that disentangles dynamics from rewards, capturing transferable motor primitives across ADL and surgical domains and supporting efficient policy generalization under embodiment differences.

(3) Through extensive simulation and real-world experiments on the dVRK, we demonstrate that ADL pretraining accelerates adaptation and yields consistently higher sample efficiency than imitation learning and reinforcement learning baselines, providing initial evidence that cross-domain motor transfer is both practical and effective for data-limited surgical settings.

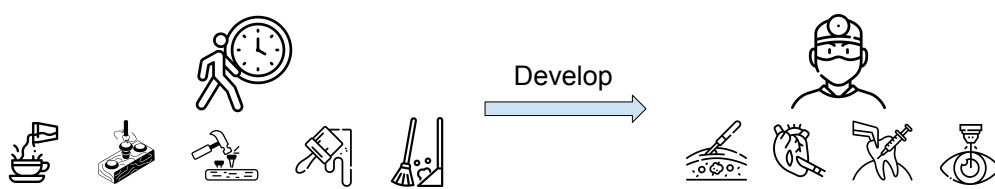

Figure 1: Expert surgical skills can be progressively developed from everyday human tool use. Daily activities such as cooking, crafting, and cleaning cultivate fundamental motor skills and tool-handling abilities, which can be transferred and refined into advanced surgical skills required for surgical procedures.

## 2 Related Works

**Structured Priors and Model-Based Control.** Classical controllers introduce strong inductive biases into surgical automation by exploiting explicit kinematic and dynamic models together with task-specific constraints, such as haptic-guided shared control (Selvaggio et al., 2019), virtual fixtures (Marinho et al., 2020), and visual needle guidance (Özgüner et al., 2021). While effective under calibrated conditions, these approaches depend heavily on hand-crafted task models and repeated parameter tuning, which restrict scalability in variable clinical environments. In contrast, our framework eliminates the need for manual modeling by learning sensorimotor primitives directly from human ADL datasets and adapting them to surgical robot task learning.

**Reinforcement Learning for Surgical Robot Task Learning.** Reinforcement learning (RL) provides a framework for training surgical policies through trial-and-error interaction without requiring expert demonstrations. Prior studies have applied RL to tissue manipulation (Shahkoo & Abin, 2023), deformable object control (Scheikl et al., 2022), ultrasound probe navigation (Bi et al., 2022), and human-in-the-loop control frameworks (Long et al., 2023). While these works demonstrate the potential of RL, allowing surgical robots to interact directly with tissue is both high-risk and highly sample-inefficient, making online RL impractical in clinical settings. In contrast, our approach reduces the need for extensive on-robot exploration by pre-training transferable motor primitives on abundant ADL datasets and then adapting them to surgical robot task learning. This shift from interaction-heavy policy search to representation-based transfer enables safer, faster, and more data-efficient task learning.

**Reward Learning Approaches.** Our work is related to reward learning in that both aim to reduce reliance on dense, hand-crafted supervision in robotic policy learning. Inverse reinforcement learning (IRL) seeks to infer reward functions from expert demonstrations, with classic formulations recovering policies through apprenticeship learning and addressing ambiguity via maximum-entropy objectives (Abbeel & Ng, 2004; Ziebart et al., 2008; Ng et al., 2000). Although these methods have advanced reward learning, their applicability to surgical robotics is constrained by the scarcity and cost of high-quality surgeon demonstrations. In contrast, our framework circumvents this dependence by extracting motor primitives from abundant non-surgical ADL datasets and adapting them to surgical tasks through lightweight fine-tuning with minimal online interaction. This positions our method as complementary to reward learning: while reward learning infers objectives from scarce demonstrations, we shift the supervision source to scalable ADL-based pretraining, directly addressing the data bottleneck in surgical robot task learning.

**Learning from Demonstrations.** Learning from Demonstrations (LfD) is a central paradigm in surgical robotics, enabling systems to imitate expert trajectories from curated datasets. Approaches range from probabilistic movement primitives (Su et al., 2021), to trajectory segmentation (Pan et al., 2022), to diffusion-based imitation of deformable tissue manipulation (Scheikl et al., 2024). Yet LfD is fundamentally constrained by the difficulty of acquiring surgical demonstrations, which are costly, logistically complex, and limited by surgeon availability (Schmidgall et al., 2024; Sridhar et al., 2017). Our framework mitigates this dependency by leveraging ADL datasets (Petrich et al., 2022; Huang & Sun, 2019), enabling surgical robot task learning to be initialized from transferable sensorimotor primitives and adapted to downstream surgical tasks with reduced reliance on expert data.

**Foundation Models.** Foundation models demonstrate that large-scale datasets can drive transferable representations for diverse downstream tasks in language and vision (Wiggins & Tejani, 2022). In robotics, such

Table 1: Comparison of different approaches for skill learning. Model-based control lacks adaptability; reinforcement learning requires extensive interaction; reward learning and imitation learning depend on task-specific data; and foundation models remain costly and surgery-specific. In contrast, our method learns transferable state–action representations from human ADL data without surgical demonstrations, enabling efficient and generalizable surgical skill learning.

| | Structured Model-Based Control | Reinforcement Learning | Reward Learning | Imitation Learning | Foundation Models | Our Method |
|---|---|---|---|---|---|---|
| Independence from in-domain data | ✓ | ✓ | × | × | ✓ | ✓ |
| No reliance on structured priors | × | ✓ | ✓ | ✓ | ✓ | ✓ |
| Interaction efficiency | NA | × | × | NA | NA | ✓ |
| Sample efficiency | NA | × | × | NA | NA | ✓ |
| Generalization capability | × | × | × | × | ✓ | ✓ |
| Computational efficiency | NA | ✓ | ✓ | ✓ | × | ✓ |

models have achieved cross-embodiment and multi-modal transfer (Reed et al., 2022; Brohan et al., 2022). However, these approaches typically require vast amounts of synchronized sensorimotor demonstrations, which are infeasible to obtain in safety-critical surgical contexts. Our work provides a scalable alternative: instead of relying on massive task-specific datasets, we exploit widely available human ADL datasets as a safe surrogate for pretraining transferable representations for surgical robot task learning.

**Representation Learning.** Central to our method is representation learning that separates environment dynamics from task rewards. Successor features (SF)(Barreto et al., 2017; 2018) provide a principled mechanism for this decomposition: dynamics are encoded via expected feature accumulations, while rewards are computed using linear weights. Extensions such as universal SFs and generalized policy improvement (GPI)(Borsa et al., 2018) support multi-task reuse and transfer under reward variation. In robotics, SF-based transfer has been studied for goal generalization (Zhang et al., 2017), but applications to surgical robotics remain limited due to data scarcity and embodiment mismatch. Successor features for cross-embodiment and cross-task transfer were previously explored in (Hu et al., 2025), where human daily activity skills were leveraged to surgical task learning. However, that approach depends on a manually curated set of "relevant" human activities and then selects a single most relevant activity from this candidate pool as the transferable policy, which limits scalability and robustness when relevance is ambiguous or shared across multiple skills. In contrast, our work eliminates this manual selection requirement by introducing a modular successor feature framework that learns multiple human skill modules and dynamically composes them during downstream surgical task learning.

To provide a holistic comparison, Table 1 contrasts existing approaches across key dimensions, where our method stands out by leveraging human ADL datasets to achieve both high generalization and low data requirements.

## 3 Background and Problem Setup

**Problem Setup.** We formulate *surgical robot task learning* as a problem of cross-embodiment and cross-task transfer from human Activities of Daily Living (ADLs). Specifically, we define two Markov Decision Processes (MDPs): a source-domain MDP for ADL behavior and a target-domain MDP for surgical robot tasks. The ADL MDP is defined as $\mathcal{M}_{\text{ADL}} = (\mathcal{S}_{\text{ADL}}, \mathcal{A}_{\text{ADL}}, P_{\text{ADL}}, r_{\text{ADL}}, \gamma_{\text{ADL}})$, where $\mathcal{S}_{\text{ADL}}$ and $\mathcal{A}_{\text{ADL}}$ represent human-centric states and actions (e.g., hand postures, tool-use motions), $P_{\text{ADL}}$ specifies the transition dynamics, and $r_{\text{ADL}}$ is a task-specific reward. The surgical MDP is defined as $\mathcal{M}_{\text{Surg}} = (\mathcal{S}_{\text{Surg}}, \mathcal{A}_{\text{Surg}}, P_{\text{Surg}}, r_{\text{Surg}}, \gamma_{\text{Surg}})$, where the state and action spaces correspond to robot kinematics and surgical control contexts.

Although the two domains differ in embodiment, observation spaces, and task objectives, we hypothesize that they share common low-level motor control structures, which refers to shared structure at the level of short-horizon transition dynamics, which is precisely the level captured by successor features. To capture this shared structure, we learn a transferable embedding function $f : (s, a) \mapsto \mathbb{R}^d$ that maps state–action pairs from ADL tasks into a latent space encoding sensorimotor dynamics. The function $f$ is trained across

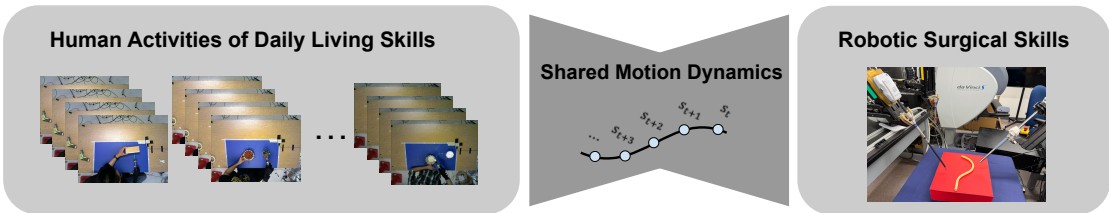

Figure 2: Illustration of our proposed framework for robotic surgical skill acquisition via motor knowledge transfer. Human ADL datasets encompass a broad spectrum of manipulation behaviors, including tool use and fine-motor tasks. Although ADL and surgical tasks differ in semantic goals and environments, they share underlying motor control structures. Our approach leverages these shared dynamics to learn transferable representations that enable efficient policy adaptation in surgical domains.

a diverse set of ADL tasks $\{\mathcal{M}^1_{\text{ADL}}, \mathcal{M}^2_{\text{ADL}}, \dots\}$, each with a different reward function $r^i_{\text{ADL}}$, such that the learned embedding generalizes across tasks with varied goals but similar motion patterns.

Our objective is to transfer this motor-centric embedding space to the surgical domain and use it to guide the learning of a surgical policy $\pi_{\text{Surg}}(a_{\text{Surg}} \mid s_{\text{Surg}})$ that maximizes expected cumulative reward in $\mathcal{M}_{\text{Surg}}$. By reusing motor knowledge distilled in $f$, the framework enables fast adaptation in data-constrained surgical settings, thereby reducing reliance on expert demonstrations and supporting sample-efficient learning from minimal in-domain interactions.

**Transfer with SF & GPI:** A central question in our formulation is whether there exists a function $f : (s, a) \mapsto \mathbb{R}^d$ that can map state-action pairs from the ADL domain into a low-dimensional latent space that captures generalizable motor dynamic patterns. If such a representation exists and can be learned from diverse ADL tasks, it opens the possibility of transferring these latent vectors to a different domain, such as surgical robotics, where the goal is to efficiently learn a control policy $\pi_{\text{Surg}}(a_{\text{Surg}} \mid s_{\text{Surg}})$ with minimal online interaction samples.

To instantiate this function $f$, we adopt the framework of *successor features* (SFs), a class of predictive representations in reinforcement learning known to support transfer across tasks with shared dynamics but differing reward structures (Momennejad, 2020; Barreto et al., 2017). SFs decompose the action-value function into a product of two components: one capturing environment dynamics and the other encoding task-specific rewards. Formally, the reward associated with a transition $(s_t, a_t, s_{t+1})$ is given by:

$$r(s_t, a_t, s_{t+1}) = \phi(s_t, a_t, s_{t+1})^\top \mathbf{w} \tag{1}$$

where $\phi(s_t, a_t, s_{t+1}) \in \mathbb{R}^d$ denotes a feature vector characterizing the transition, and $\mathbf{w} \in \mathbb{R}^d$ are task-specific weights. These features $\phi$ reflect motor dynamics that are agnostic to task semantics and thus provide the structure necessary for transfer.

The state-action value function under policy $\pi$ can be rewritten as:

$$Q^\pi(s_t, a_t) = \mathbb{E}^\pi \left[ \sum_{i=t}^{\infty} \gamma^{i-t} r(s_i, a_i) \mid S_t = s_t, A_t = a_t \right] = \psi^\pi(s_t, a_t)^\top \mathbf{w} \tag{2}$$

where $\psi^\pi(s_t, a_t) = \mathbb{E}_\pi \left[ \sum_{i=0}^{\infty} \gamma^i \phi(s_{t+i}, a_{t+i}, s_{t+i+1}) \right]$ is the *successor feature* corresponding to $(s_t, a_t)$ under policy $\pi$. The vector $\psi^\pi$ encodes expected discounted dynamics and is thus invariant to task-specific reward structures.

The standard formulation of successor features relies on the assumption that the reward function is a linear combination of the feature vector, i.e., $r(s_t, a_t, s_{t+1}) = \phi(s_t, a_t, s_{t+1})^\top \mathbf{w}$. While this linearity simplifies transfer, it may be restrictive in complex settings like robotic surgery. To overcome this, *deep successor features* (DSFs) (Barreto et al., 2018) extend the original formulation by learning nonlinear feature embeddings via deep neural networks, enabling richer representations of motor dynamics.

In our framework, successor features are learned from a diverse set of ADL tasks $\{\mathcal{M}^i_{\text{ADL}}\}$, each differing in reward function but sharing similar underlying dynamics. These learned features can be reused in the surgical target domain $\mathcal{M}_{\text{Surg}}$ using *Generalized Policy Improvement* (GPI) (Barreto et al., 2018). Given a set of source policies $\{\pi_i\}_{i=1}^{n_{\text{train}}}$ with their corresponding successor features $\{\psi^{\pi_i}(s,a)\}$, the optimal action in a novel surgical task with reward weights $w_{\text{test}}$ is selected by:

$$\pi(s_t; w_{\text{test}}) = \arg\max_{a \in \mathcal{A}} \max_{i \in \{1,\ldots,n_{\text{train}}\}} \left\{ \psi^{\pi_i}_t(s_t, a)^\top w_{\text{test}} \right\} \tag{3}$$

This policy selection strategy enables the surgical robot to generalize across tasks by evaluating candidate actions under previously learned source policies and selecting the one predicted to perform best. In doing so, the robot can adapt rapidly to new surgical scenarios by leveraging latent motor knowledge distilled from ADL datasets, without relying on extensive task-specific demonstrations. This establishes a scalable and sample-efficient pathway for reusing human ADL task dataset in data-constrained surgical robot task learning.

**Challenges.** While successor features (SFs) have demonstrated effectiveness in multi-task transfer, prior applications have largely been restricted to consistent embodiment settings. Transferring motor knowledge from human ADL datasets to robotic surgical tasks introduces two key challenges: embodiment mismatch and reward uncertainty.

First, $\mathcal{M}_{\text{ADL}}$ and $\mathcal{M}_{\text{Surg}}$ differ fundamentally in embodiment. Human motion is high-dimensional, flexible, and enriched by proprioceptive feedback, whereas surgical robots operate in constrained workspaces with task-specific end-effectors. As a result, SFs $\psi^\pi_{\text{ADL}}$ learned from $\mathcal{D}_{\text{ADL}}$ may not directly generalize to $\mathcal{S}_{\text{Surg}}, \mathcal{A}_{\text{Surg}}$ due to representational misalignment.

Second, unlike standard SF+GPI settings where the reward weights $w_{\text{test}}$ of the target task are assumed to be known, robotic surgical tasks only provide the task rewards $r_{\text{Surg}}$. Without explicit knowledge of a reward vector, directly applying GPI to construct high-performing surgical policies becomes infeasible. Furthermore, in $\mathcal{M}_{\text{Surg}}$, selecting which source ADL tasks $\mathcal{M}^i_{\text{ADL}}$ to reuse remains a non-trivial problem.

## 4 Method

To address the challenges of cross-embodiment and cross-task transfer from human ADL domains $\{\mathcal{M}^i_{\text{ADL}}\}$ to surgical robot tasks $\mathcal{M}_{\text{Surg}}$, we decompose the problem into three key components: representation learning, skill composition, and reward adaptation. Together, these components establish a modular successor feature (SF) framework (Carvalho et al., 2023) that enables transferable state–action representations and efficient policy adaptation.

1. **Representation Learning.** We first construct a shared latent space $\mathcal{Z} \subset \mathbb{R}^d$ by learning deep successor features (SFs) $\psi^{\pi_i}(s,a)$ from a set of source ADL tasks $\{\mathcal{M}^i_{\text{ADL}}\}$. Each value function factorizes as $Q^{\pi_i}(s,a) = \psi^{\pi_i}(s,a)^\top w^i$, where $w^i \in \mathbb{R}^d$ are task-specific reward weights and $\phi(s,a,s')$ is a shared transition feature embedding. This decoupling isolates transferable dynamics in $\psi$, allowing reuse across ADL tasks with different rewards $r^i_{\text{ADL}}$ and providing a foundation for cross-domain transfer.

2. **Skill Composition.** Each learned SF $\psi^{\pi_i}$ captures a reusable motor primitive from a source task $\mathcal{M}^i_{\text{ADL}}$. To transfer these primitives to $\mathcal{M}_{\text{Surg}}$, we aggregate $\{\psi^{\pi_i}\}$ using a learned attention mechanism that prioritizes the source features most relevant to the surgical domain. This composition produces a set of candidate policies aligned with surgical dynamics, enabling flexible adaptation without retraining from scratch.

3. **Reward Adaptation.** With limited interaction data $\mathcal{D}_{\text{Surg}}$, we estimate the reward vector $w_{\text{Surg}}$ for $\mathcal{M}_{\text{Surg}}$ via supervised regression on $(s,a,r)$ tuples. Equipped with $w_{\text{Surg}}$, we apply Generalized Policy Improvement (GPI) to synthesize an adapted surgical policy:

$$\pi(s) = \arg\max_{a \in \mathcal{A}_{\text{Surg}}} \max_i \psi^{\pi_i}(s,a)^\top w_{\text{Surg}}. \tag{4}$$

This mechanism leverages transferable motor structure in $\mathcal{Z}$ to achieve rapid and data-efficient adaptation, bypassing the need for extensive task-specific demonstrations.

This modular SF framework allows compositional transfer of motor knowledge distilled from diverse human tool-use behaviors, providing a principled pathway to scalable and data-efficient surgical robot task learning. In this work, the human daily activity dataset is required to contain motor-level information, specifically the translation and orientation of the tool or human hand.

## 4.1  Offline Training of Successor Features on Human ADL Data

To enable cross-domain transfer to surgical robot tasks, we first learn transferable motor representations from a diverse set of human ADL tasks $\{\mathcal{M}^i_{\text{ADL}}\}^n_{i=1}$. Each task is defined as an MDP $\mathcal{M}^i_{\text{ADL}} = (\mathcal{S}^i, \mathcal{A}^i, P^i, r^i, \gamma)$, where the reward function $r^i$ encodes a distinct motor objective. Our goal is to distill task-agnostic motor dynamics in the form of successor features (SFs), while learning task-specific reward weights $\{w^i_{\text{ADL}}\}$ that enable downstream adaptation to $\mathcal{M}_{\text{Surg}}$.

We adopt a modular successor feature framework (Carvalho et al., 2023), where each module $k \in \{1, \ldots, n\}$ captures a reusable motor primitive derived from one ADL task. At time $t$, the latent state for module $k$ is computed from the current observation $x^k_t$ and recurrent hidden state $s^k_{t-1}$:

$$s^k_t = s_{\theta_k}(x^k_t, s^k_{t-1}), \tag{5}$$

where $\theta_k$ denotes module-specific parameters.

For each skill module, we define a cumulant function $\phi^k(s, a, s') \in \mathbb{R}^d$ and corresponding successor features $\psi^k(s, a) \in \mathbb{R}^d$ with respect to its reward weights $w^k_{\text{ADL}}$:

$$\phi^k_t = \phi_\theta(s^k_t, a_t, s^k_{t+1}), \quad \psi^k_t = \psi_\theta(s^k_t, a_t). \tag{6}$$

The successor features encode the expected discounted future cumulants under the behavior policy $\pi$:

$$\psi^k(s^k_t, a_t) = \mathbb{E}_\pi \left[ \sum_{i=0}^{\infty} \gamma^i \phi^k_{t+i} \right]. \tag{7}$$

To enable skill composition across ADL modules, we construct joint representations by concatenating module-level cumulants and SFs:

$$\phi_t = \left[ \phi^1_t, \ldots, \phi^n_t \right], \quad \psi_t = \left[ \psi^1_t, \ldots, \psi^n_t \right]. \tag{8}$$

This modular construction allows transferable motor primitives from heterogeneous ADL tasks to be combined into a unified predictive representation, providing the basis for subsequent transfer to surgical domains.

## 4.2  Surgical Task Learning: Policy Inference via Generalized Policy Improvement (GPI)

Each module $k$ captures a distinct aspect of human motor behavior, such as grasping, reaching, or object manipulation. The associated cumulants $\phi^k(s, a)$ represent immediate skill-specific signals. In the surgical domain, we approximate the reward function as a linear combination of these cumulants, with module-specific weights $w^k_{\text{Surg}}$ learned from surgical data. This formulation enables the model to balance the contributions of individual motor primitives when adapting to the target task.

We estimate the reward weights $w^*_{\text{Surg}}$ by solving the following regression problem:

$$w^*_{\text{Surg}} = \arg\min_w \mathbb{E}_{(s,a,r) \sim \mathcal{D}_{\text{Surg}}} \left[ \left( r_{\text{Surg}} - \sum_{k=1}^{K} \phi^k(s, a) w^k \right)^2 \right], \tag{9}$$

where $\mathcal{D}_{\text{Surg}}$ denotes the dataset of state–action–reward tuples collected in the surgical environment, and $w^k$ scales the contribution of cumulant $\phi^k(s, a)$ from the $k$-th ADL module. This regression process effectively aligns ADL-derived motion dynamics with the reward structure of the surgical task.

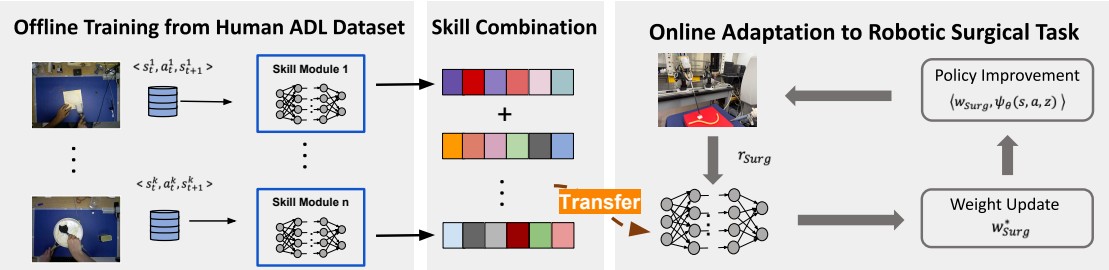

Figure 3: Overview of the proposed framework. Human Activities of Daily Living (ADL) datasets (e.g., DIM dataset (Huang & Sun, 2019)) are first used for offline training, where deep successor features are learned to construct a predictive state–action representation space. Each learned module captures transferable motor dynamics across diverse tool-use behaviors. In the skill transfer phase, these modules are combined and adapted to surgical domains, where online updates of reward weights enable efficient policy improvement. This process allows surgical robots to leverage motor knowledge from ADL data for fast, scalable, and data-efficient task learning.

Given the estimated weights $w_{\text{Surg}}$, we sample $N$ candidate actions from the continuous action space and derive the adapted surgical policy via Generalized Policy Improvement (GPI):

$$\pi(s) = \arg\max_{a \in \mathcal{A}_{\text{Surg}}} \max_{z \in \mathbb{M}_{\text{ADL}}} \left\{ \psi_\theta(s, a, z)^\top w_{\text{Surg}} \right\}, \tag{10}$$

where $z$ indexes the set of source policies or task vectors $\mathbb{M}_{\text{ADL}}$, and $\psi_\theta(s, a, z)$ denotes the successor features of the $z$-th source module, evaluated at $(s, a)$.

This maximization selects the source task representation $z$ that best aligns with the surgical reward structure and then chooses the action $a$ that maximizes expected return. In this way, the surgical robot reuses task-agnostic features and motor primitives distilled from ADL datasets, while adapting them to the specific reward function of $\mathcal{M}_{\text{Surg}}$. This approach enables efficient policy transfer with minimal surgical supervision.

## 5 Theoretical Analysis

We now present a theoretical analysis of transferring state–action representations from human ADL tasks to surgical domains. Our goal is to establish performance guarantees for the resulting surgical policy $\pi_{\text{Surg}}$ under the assumption of a shared feature space between source and target tasks.

**Theorem 1:** Assume the state–action space $\mathcal{S} \times \mathcal{A}$ is finite. There exists a distribution $\rho$ on $\mathcal{S} \times \mathcal{A}$ with $\rho(s, a) \geq \rho_{\min} > 0$ for all $(s, a)$. Rewards are bounded by $|r_{\text{Surg}}(s, a)| \leq R$ and $||w||_2 \leq W$. Let $\{\phi^k(s, a)\}_{k=1}^K$ be $K$ feature functions and define $\phi(s, a) := \left[ \phi^1(s, a), \ldots, \phi^K(s, a) \right] \in \mathbb{R}^K$, with $||\phi(s, a)||_2 \leq B$ for all $(s, a)$ and $d = K * Nmodule$. We approximate the surgical reward with a linear model $\hat{r}_{\text{Surg}}(s, a; w) = \phi(s, a)^\top w$, $w \in \mathbb{R}^K$.

Given a dataset $\mathcal{D} = \{(s_i, a_i, r_i)\}_{i=1}^n$ from the surgical task, let

$$\hat{w} = \arg\min_w \ \mathbb{E}_{(s,a,r) \sim \mathcal{D}} \left[ \left( r_{\text{Surg}}(s, a) - \sum_{k=1}^K \phi^k(s, a) w^k \right)^2 \right].$$

For any policy $\pi$, let $Q^\pi$ denote the true action–value function under $r_{\text{Surg}}$, and let $\hat{Q}^\pi$ denote the action–value function computed under the approximated reward $\hat{r}_{\text{Surg}}(\cdot, \cdot; \hat{w})$. Then, we have at least $1 - \delta$ possibility that

$$||Q_\pi - \hat{Q}_\pi(\hat{w})||_\infty \leq \frac{1}{(1 - \gamma)\sqrt{\rho_{\min}}} \sqrt{\inf_w \mathbb{E}\left[ \left( r_{\text{Sur}}(s, a) - \phi(s, a)^\top w \right)^2 \right] + C (BR + BW)^2 \sqrt{\frac{d + \log(1/\delta)}{n}}}.$$

Theorem 1 establishes that, provided the ADL-derived feature space is sufficiently expressive to approximate the surgical reward function, the induced value function uniformly approximates the true value function. As the richness of the feature class and the size of the ADL dataset increase, the approximation error vanishes up to inherent statistical limits. This result formalizes the conditions under which ADL-derived representations yield consistent and accurate policy evaluation for surgical tasks. A detailed proof is provided in Appendix B.1.

**Theorem 2:** Under the assumptions of Theorem 1, let $\Pi = \{\pi_1, \pi_2, \ldots, \pi_M\}$ be a finite set of stationary policies. For each $\pi \in \Pi$, let $Q^\pi$ denote the true action–value function under $r_{\text{Surg}}$, and let $\hat{Q}^\pi$ denote the action–value function computed under the approximated reward $\hat{r}_{\text{Surg}}(\cdot, \cdot; \hat{w}) = \phi(\cdot, \cdot)^\top \hat{w}$.

Then, with probability at least $1 - \delta$, uniformly for all $\pi \in \Pi$,

$$||Q_\pi - \hat{Q}_\pi(\hat{w})||_\infty \leq \frac{1}{(1-\gamma)\sqrt{\rho_{\min}}}\sqrt{\inf_w \mathbb{E}\left[\left(r_{\text{Sur}} - \phi^\top w\right)^2\right] + C\,(BR + BW)^2 \sqrt{\frac{d + \log(1/\delta)}{n}}}.$$

Define a new policy $\pi'$ such that $\pi'(s) \in \arg\max_a \max_{\pi \in \Pi} \hat{Q}_\pi(s, a)$. Then, for all $(s, a)$,

$$Q_{\pi'} \geq \max_{\pi \in \Pi} Q_\pi \; - \; \frac{2}{(1-\gamma)^2 \sqrt{\rho_{\min}}}\sqrt{\inf_w \mathbb{E}\left[\left(r_{\text{Sur}} - \phi^\top w\right)^2\right] + C\,(BR + BW)^2 \sqrt{\frac{d + \log(1/\delta)}{n}}}).$$

Theorem 2 further establishes that, even when the reward function is only approximately recovered from ADL-derived features, the resulting policy $\pi'$ achieves value close to that of the best policy in the hypothesis class $\Pi$. The performance gap decreases as the dataset size grows and the feature space becomes more expressive, with guarantees holding uniformly over all $\pi \in \Pi$. This result provides a formal performance bound for policy transfer under approximate reward representations. A full proof is provided in Appendix B.2.

# 6 Experiment

In this section, we validated the effectiveness of transfering from human ADL dataset in robotic surgical task learning from both simulation and real-world experiments.

## 6.1 Experiment Setup

**Human ADL Dataset.** We employ the Daily Interactive Manipulation (DIM) dataset (Huang & Sun, 2019) as the source of human Activities of Daily Living (ADL) demonstrations. The DIM dataset contains 1,603 motion trials across 32 fine-grained manipulation tasks commonly observed in everyday life, such as brushing powder across a tray, unlocking a lock with a key, and filing wood.

From this dataset, we selected three representative ADL tasks that exhibit tool-use behaviors and coordinated hand–eye control, serving as foundational modules for transfer to robotic surgical tasks: (1) spearing an object using a fork, (2) lifting food with a spoon, and (3) scooping and pouring with a measuring cup.

These tasks were chosen because they emphasize precision, stability, and force control—sensorimotor attributes closely aligned with surgical manipulation, making them ideal for learning transferable motor primitives applicable to surgical robot task learning. The detail of processing DIM dataset in our experiment can be found in Appendix C.1.

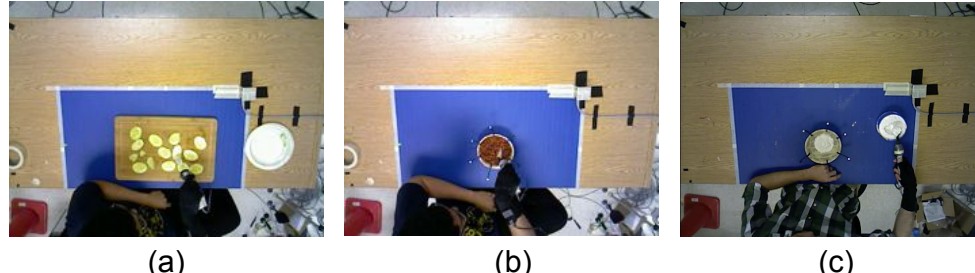

(a)          (b)          (c)

Figure 4: Examples from the Daily Interactive Manipulation (DIM) dataset illustrating three representative human Activities of Daily Living (ADL) tasks: (a) spearing an object with a fork, (b) lifting food with a spoon, and (c) scooping and pouring with a measuring cup. These fine-motor tasks are used to construct modular motor representations that serve as transferable building blocks for robotic surgical task learning on the PSM and ECM systems.

**Surgical Robot Platform.** Experiments are conducted using the *da Vinci Research Kit (dVRK)*, a clinically derived robotic platform for minimally invasive surgery that enables high-precision teleoperation with articulated robotic arms and stereo visual feedback. The system comprises two primary manipulators: the *Patient Side Manipulator (PSM)*, responsible for executing tool-based surgical actions, and the *Endoscopic Camera Manipulator (ECM)*, which provides real-time stereo imaging and viewpoint control. In our study, both PSM and ECM subsystems are utilized to evaluate cross-domain transfer and data-efficient policy learning in surgical robot task learning.

Transfering from human ADL dataset in robotic surgical task learning, we construct the transferable state-action reresentation space to overcome the cross-embiodment and cross-task gap between the human daily actvitites and surgical robot task. The core mechanism is that the shared environment dynamics extracted by the modular successor feature framework. The detail of construction of transferable state-action space can be found in Appendix D.1. It is noted that, we use the motor level information of DIM dataset, instead of the visual image, in transfer learning. The huge gap between teh human daily life and robotic surgical task will greatly increase the online interaction demand, may like training from scratch. Moreover, in our work, we assume the shared motion dynamics between the human daily activities and robotic surgical tasks, that is the probability of the next state given the current state and action when excuting one task. Hence, the observation and action spaces are low-dimensional robot or object states.

### 6.2 Simulation Experiment

**Experiment Simulator.** The simulation experiments in this study are *not* intended for sim-to-real transfer. Instead, simulation is employed to enable extensive and controlled quantitative evaluations of learning performance, scalability, and ablation results. All experiments are conducted using the open-source *SurRoL* simulator (Xu et al., 2021), a reinforcement learning–oriented platform fully compatible with the *da Vinci Research Kit (dVRK)* (Kazanzides et al., 2014). The simulator provides a reproducible environment for evaluating policy adaptation across diverse surgical task settings under varying data constraints.

Three representative surgical tasks are selected for evaluation: (1) **PSM Reaching Gauze**, where the goal is to position the PSM jaw tip precisely above a randomly placed suture gauze; (2) **ECM Static Tracking**, which requires the ECM to center a stationary target cube within the stereo camera's image space; and (3) **ECM Active Tracking**, where the ECM continuously follows a moving target cube along a dynamically generated trajectory at constant velocity.

The experimental setup of these simulation tasks can be found in Appendix C.2. These tasks collectively test both manipulation and visual tracking capabilities, providing a systematic benchmark for assessing cross-domain transfer and data efficiency in surgical robot task learning.

**Baselines.** We compare our framework against three representative baselines to assess the benefits of ADL-based pretraining and successor feature transfer: (1) **COMBO with ADL** (Yu et al., 2021), an offline model-based RL approach that pre-trains a policy on the human ADL dataset and directly deploys it to the surgical task via imitation learning. This baseline measures how well direct policy transfer performs without

explicit representation adaptation. (2) **MBPO** (Janner et al., 2019), which trains a policy from scratch on the surgical task using model-based policy optimization. It serves as a lower-bound baseline, isolating the effect of learning without any external priors. (3) **MBPO with ADL**, which initializes the policy with ADL-pretrained weights and fine-tunes it on the surgical task. This baseline evaluates the effectiveness of naive weight-level transfer, contrasting it with our proposed structured representation transfer via successor features. The training detail of the baseline algorithm and our method can be found in Appendix D.2.

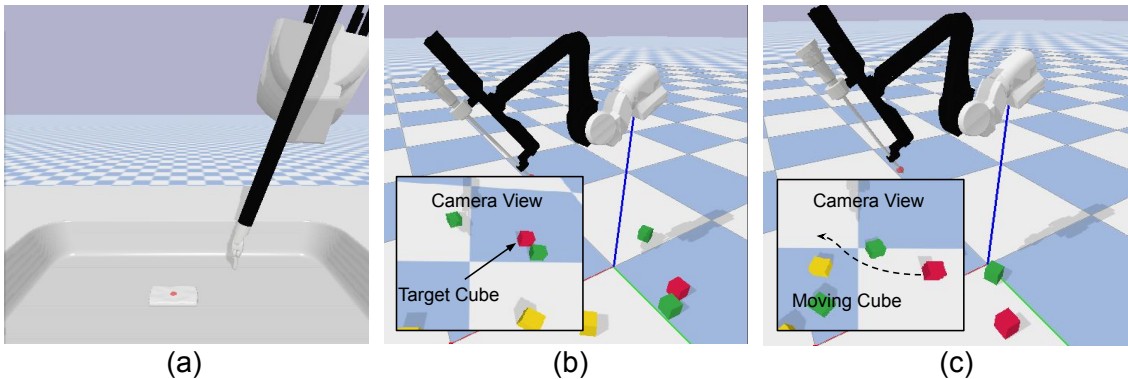

Figure 5: Simulation-based surgical tasks used for evaluation. All experiments are conducted using the open-source *SurRoL* simulator (Xu et al., 2021), a reinforcement learning–oriented platform fully compatible with the *da Vinci Research Kit (dVRK)* (Kazanzides et al., 2014). (a) **PSM Reaching Gauze**: models precise instrument positioning, simulating tool approach toward tissue or suturing material. (b) **ECM Localizing a Static Cube**: emulates maintaining a stable endoscopic view of a stationary anatomical structure. (c) **ECM Tracking a Moving Cube**: replicates dynamic camera adjustment required to follow moving or deforming surgical targets. Together, these tasks capture complementary aspects of surgical coordination—fine tool manipulation and visual stabilization—within a controlled and reproducible simulation setting.

### 6.2.1 Experiment Results

The simulation results are summarized in Fig. 6. A key observation is that naive pretraining or direct transfer of ADL policies does not suffice: the COMBO baseline, which deploys an ADL-pretrained policy without adaptation, consistently fails due to distribution mismatch between daily motions and surgical contexts. Similarly, MBPO with ADL pretraining underperforms compared to training from scratch, underscoring that weight-level initialization alone introduces harmful biases when embodiment and task dynamics differ. These negative results highlight the necessity of a structured transfer mechanism rather than undermining the value of ADL data.

In contrast, our method leverages modular successor features to systematically align ADL-derived motor primitives with surgical rewards. This results in consistently superior sample efficiency and rapid convergence across all tasks. For instance, in both PSM Reaching Gauze and ECM Localizing tasks, our approach reaches near-optimal performance in a fraction of the training episodes required by MBPO. Notably, in the ECM Tracking task, which requires dynamic adaptation, our method achieves robust performance with significantly lower variance, reflecting stable transfer and effective reuse of prior knowledge.

Although MBPO eventually approaches high success rates given sufficient interaction, such asymptotic convergence is impractical in data-limited surgical settings. Our framework, by contrast, demonstrates both fast adaptation and reduced variance, establishing its utility as a safe and scalable approach to surgical robot task learning.

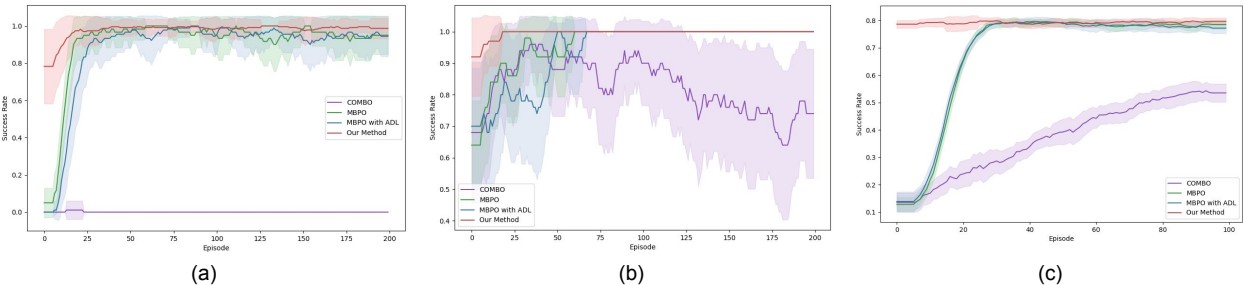

Figure 6: Simulation results across three surgical tasks: (a) **PSM Reaching Gauze**, (b) **ECM Localizing a Static Cube**, and (c) **ECM Tracking a Moving Cube**. Baselines illustrate the challenges of using ADL data without structured transfer: COMBO (direct deployment of ADL-pretrained policies) fails due to severe distribution mismatch, while MBPO with ADL pretraining performs worse than scratch training, highlighting the pitfalls of naive weight initialization. In contrast, our method leverages modular successor features to bridge embodiment and task differences, yielding consistently faster adaptation, higher sample efficiency, and lower variance. Notably, performance remains robust in dynamic tasks such as ECM Tracking, demonstrating stable and scalable transfer of ADL-derived motor knowledge to surgical domains.

To further evaluate surgical task learning, we analyze three complementary metrics: (1) **Convergence speed**, defined as the number of episodes required to reach stable success; (2) **Data efficiency**, measured by the number of environment interactions needed to attain and sustain stable performance; and (3) **Jumpstart performance**, quantified as the average success rate over the first five episodes (Tables 2).

For (1), our method achieves up to a $13.5\times$ improvement over the strongest baseline (MBPO), demonstrating rapid convergence and effective initialization in data-limited surgical settings. For (2), it consistently requires fewer environment interactions across all tasks, indicating efficient use of limited surgical experience and robust sample efficiency. For (3), our approach attains a success rate of 0.88 on the PSM task within the first few episodes—far exceeding baselines that struggle to achieve any meaningful early performance, particularly those relying on naive ADL pretraining.

While the large relative improvements in early-stage performance arise partly because baseline methods exhibit minimal initial success, this gap precisely illustrates their inability to initialize meaningful control without structured transfer. In contrast, our framework provides well-aligned representations that enable rapid adaptation from the outset, demonstrating both the necessity and effectiveness of transferring structured motor knowledge from human ADL data. Overall, these results confirm that our approach accelerates learning, enhances initial competence, and achieves safe, scalable, and data-efficient surgical robot task learning.

## 6.3 Real-world Experiment

### 6.3.1 Experiment Platform Setup

To evaluate the practical effectiveness and robustness of the proposed framework beyond controlled simulation, we conduct real-world experiments using the *da Vinci Research Kit* (dVRK)—an open research platform derived from the clinically deployed *da Vinci Surgical System* widely used in minimally invasive surgery (MIS). Real-world validation is essential, as it introduces physical constraints, actuation noise, and vision–motion coupling effects that cannot be fully modeled in simulation, thereby testing the framework's transferability under authentic robotic operating conditions.

Importantly, our experiments do not involve sim-to-real transfer; instead, we directly deploy policies on the physical robot after offline pretraining on human ADL datasets, followed by limited on-robot reinforcement learning for task adaptation. This setup allows us to isolate the framework's capability for safe, data-efficient adaptation without the confounding effects of simulator–reality mismatch.

The dVRK system is operated via open-source electronics and software (Kazanzides et al., 2014), integrated with ROS for low-latency pose-level control of its manipulators. We evaluate two core capabilities represen-

Table 2: Comparison of (1) **Convergence speed**, (2) **Data efficiency**, and (3) **Jumpstart performance** across three surgical tasks: PSM Reaching, ECM Static Tracking, and ECM Active Tracking. Convergence and data efficiency are reported as the number of episodes and environment interactions required to reach stable success, respectively, while jumpstart measures the average success rate over the first five episodes. Our method achieves up to a 13.5× improvement in convergence and sample efficiency over the strongest baseline (MBPO) and substantially higher early-stage performance, demonstrating faster, safer, and more data-efficient adaptation of human ADL knowledge to surgical tasks.

| Task | Metric | MBPO | MBPO+ADL | COMBO | Ours | Imp.(x) |
|------|--------|------|----------|-------|------|---------|
| PSM | Conv. | 34.00 | 52.00 | 200.00 | 9.00 | 3.80 |
| | Data | 7381.00 | 9573.00 | 69998.00 | 849.00 | 8.70 |
| | Jump. | 0.05 | 0.00 | 0.00 | 0.88 | 17.60 |
| ECMStat. | Conv. | 55.00 | 45.00 | 200.00 | 13.00 | 3.50 |
| | Data | 7317.00 | 6985.00 | 21294.00 | 1659.00 | 4.40 |
| | Jump. | 0.64 | 0.70 | 0.68 | 0.92 | 1.30 |
| ECMAct. | Conv. | 27.00 | 34.00 | 100.00 | 2.00 | 13.50 |
| | Data | 21600.00 | 27200.00 | 80000.00 | 1600.00 | 13.50 |
| | Jump. | 0.13 | 0.14 | 0.14 | 0.79 | 5.70 |

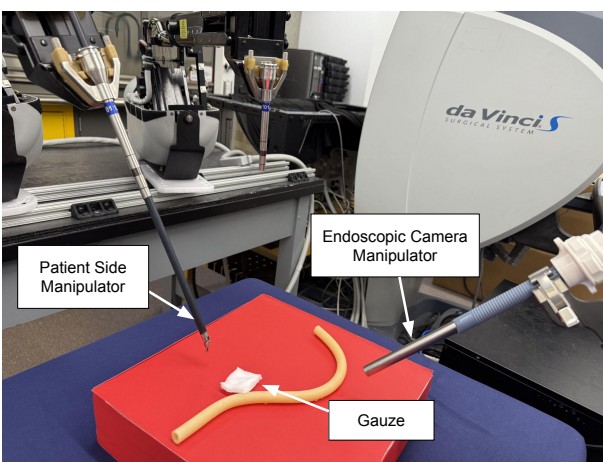

Figure 7: Real-world robotic experiment platform built on the *da Vinci Research Kit* (dVRK). The Patient Side Manipulator (PSM) performs precise tool manipulation, while the Endoscopic Camera Manipulator (ECM) regulates the stereo endoscopic view. This integrated system provides a realistic testbed for evaluating the proposed framework under clinically relevant conditions, capturing the sensorimotor control between surgical instruments and the endoscopic camera.

tative of surgical dexterity: (1) precise tool approach using the **Patient Side Manipulator (PSM)** and (2) stable surgical field visualization using the **Endoscopic Camera Manipulator (ECM)**. The complete experimental platform is shown in Fig. 7, encompassing both instrument manipulation and camera control under clinically realistic conditions.

### 6.3.2 Experiment Task Setting

In the real-world evaluation, we focus on three representative tasks designed to capture essential aspects of surgical coordination. The **PSM gauze-reaching** task (Fig. 8(a)) requires the robot to reach randomly placed targets on a surgical plane, assessing precise instrument positioning and trajectory control under real kinematic constraints.

The **ECM static tracking** task (Fig. 8(b)) involves maintaining a stationary target at the center of the endoscopic view, reflecting the need for visual stability during fine manipulation. The **ECM active tracking**

task (Fig. 8(c)) requires continuous tracking of a moving target, modeling dynamic camera adjustments encountered when anatomical structures shift or deform. The experimental setup of these real-world tasks can be found in Appendix C.3. Together, these tasks span both PSM precise manipulation and ECM stable surgical view camera control for evaluating the effectiveness and robustness of our method under practical surgical conditions.

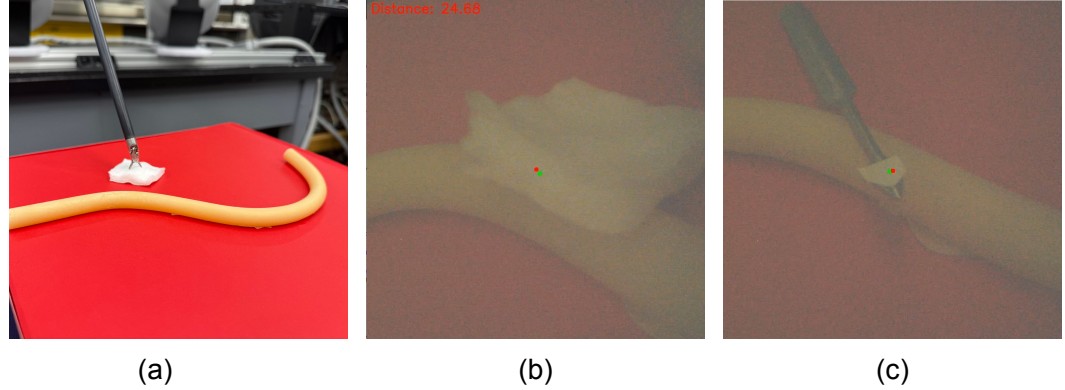

(a)            (b)            (c)

Figure 8: Real-world experiment setup: (a) **PSM Reaching Gauze**, (b) **ECM Localizing Gauze**, and (c) **ECM Tracking a Moving Tool**. Each surgical task is learned by transferring motor representations pretrained on human ADL datasets, which include daily activities such as using a spoon to pick up objects, spearing with a fork, and scooping and pouring with a measuring cup. These everyday tool-use behaviors capture transferable visuomotor primitives—grasping, reaching, and tracking—that serve as the foundation for cross-domain adaptation in robotic surgical task learning.

### 6.3.3 Experiment Results

The first set of real-world experiments evaluates the **PSM Reaching Gauze** task, which emulates fine-grained instrument positioning analogous to targeting soft tissue or grasping surgical gauze. As shown in Fig. 9, the proposed method enables the PSM to consistently reach the designated target region with progressively improved efficiency across repeated trials. Target positions were uniformly randomized across episodes to eliminate positional bias, ensuring that the observed reduction in step count reflects genuine policy improvement rather than task simplification. Due to the limited and non-uniform coverage of the state–action space in the ADL dataset, the transferred representations generalize unevenly across different regions of the task space, resulting in performance variability across targets rather than a strictly monotonic trend. The overall trend reflects a gradual reduction in required steps as adaptation progresses, indicating improved task efficiency despite local fluctuations. As the testing number of the target gauze increases, we expect these local fluctuations to average out, revealing a clearer overall downward trend that reflects improving task efficiency during adaptation.

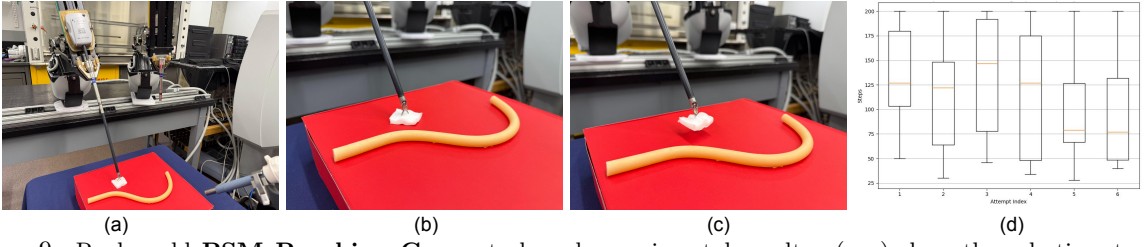

(a)            (b)            (c)            (d)

Figure 9: Real-world **PSM Reaching Gauze** task and experimental results. (a–c) show the robotic setup and sample trajectories; (d) presents the step distribution across six training attempts. Success is defined as the PSM jaw tip reaching within 3 mm of a randomly placed gauze center, simulating fine tool targeting during surgical manipulation. Target locations were uniformly randomized across trials to prevent positional bias. The consistent downward trend in step count across episodes reflects genuine policy refinement and improved motion efficiency, confirming stable adaptation of ADL-pretrained representations to real-robot control.

We next evaluate ECM-based camera control in two representative settings. In the **static gauze tracking** task, the endoscopic camera must maintain a stationary gauze centered in its visual field, analogous to stabilizing the surgical workspace during delicate procedures. As shown in Figure 10, the pixel-space distance between the gauze centroid and the image center decreases consistently across trials, indicating improved visual–motor coordination. This trend confirms that the policy effectively transfers visuomotor priors from human ADL data—such as gaze–hand coupling and smooth pursuit control (Feng et al., 2023)—while requiring only minimal online fine-tuning.

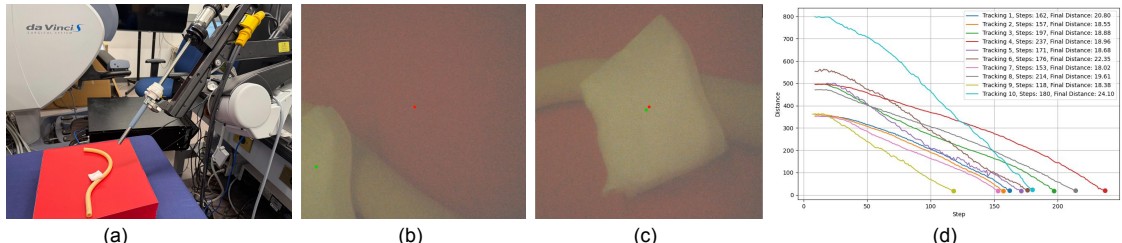

(a)            (b)            (c)            (d)

Figure 10: ECM tracking gauze experiment platform and results. (a) Experimental setup with the ECM-mounted stereo endoscope observing a randomly placed gauze. (b–c) Visual feedback examples showing the image-space centroid of the gauze (green) relative to the camera center (red). (d) Quantitative evaluation over ten tracking trials, where the metric is the pixel-space Euclidean distance between the gauze centroid and image center. The consistent downward trend across trials indicates that the ECM learns to stabilize and re-center the target more efficiently over time, even under natural lighting variations.

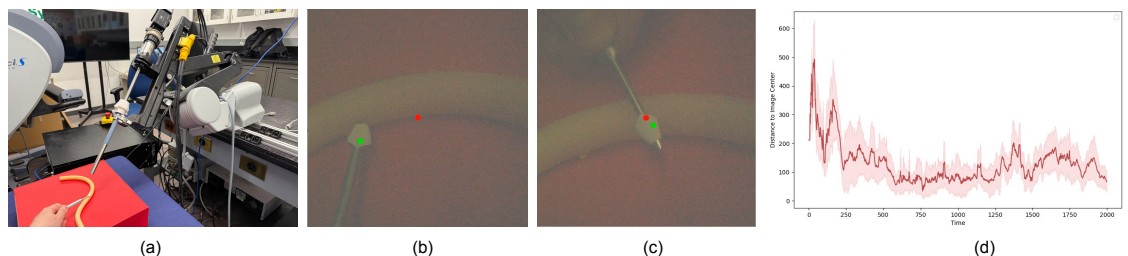

(a)            (b)            (c)            (d)

Figure 11: ECM dynamic tracking of a moving surgical tool under real-world conditions. In this task, the ECM autonomously adjusts the endoscopic view to keep a hand-manipulated surgical instrument centered in the image frame. Performance is quantified by the normalized pixel distance between the instrument centroid and the image center (averaged per frame), with shaded regions indicating standard deviation over three independent runs. The consistent downward trend confirms that tracking accuracy improves over time despite variability in human tool motion, demonstrating robust visual–motor adaptation without any sim-to-real transfer or scripted trajectories.

The **dynamic tracking** experiment (Figure 11) evaluates adaptive endoscope control under real, non-scripted motion. In this setting, the ECM autonomously tracks a surgical instrument manipulated by a human operator, emulating realistic multi-arm surgical coordination. Despite inherent variability in human tool motion, the policy consistently reduces tracking error across trials, indicating robust visual–motor adaptation and stable camera centering behavior. Importantly, this improvement arises from direct real-world learning, without any sim-to-real transfer, demonstrating that ADL-pretrained visuomotor representations generalize effectively to dynamic, human-in-the-loop surgical scenarios. Together, these results confirm that the proposed framework supports both static stabilization and real-time adaptive camera control, two essential capabilities for autonomous surgical assistance.

In summary, the real-world experiments validate the proposed framework's ability to transfer human motor knowledge—originally learned from ADL datasets—to robot-assisted surgical tasks. Our method enables both precise tool manipulation, as demonstrated in the PSM reaching experiments, and stable visual control in static and dynamic ECM tracking scenarios. These results confirm that ADL-derived visuomotor representations generalize effectively from everyday tool-use behaviors to clinically relevant surgical motions, without requiring extensive in-domain demonstrations or sim-to-real fine-tuning. Together, these capabilities

fulfill two fundamental prerequisites for surgical autonomy: accurate instrument positioning and adaptive visual feedback, ensuring safe, data-efficient, and scalable robotic operation.

# 7 Ablation Study

In this section, we analyze the learned representation dynamics through ablation studies, examining how modular components evolve and interact during downstream surgical task learning. We further investigate how the decomposition and recomposition of these modules affect representation quality, transfer efficiency, and overall performance in robotic surgical task learning.

## 7.1 Activation Dynamics of ADL Skill Modules During Surgical Tasks

To interpret how human-derived motor knowledge is reused during surgical task inference, we analyze the *activation dynamics* of individual ADL skill modules—defined as each module's normalized contribution to the predicted value estimate $\psi(s, a)$—across three representative surgical tasks (Figure 12). Module 1, Module 2, and Module 3 are pretrained respectively on *spoon–pick*, *cup–scoop/pour*, and *fork–spear* behaviors. Subfigures (a), (b), and (c) correspond to different target configurations within the same surgical task, such as gauze or cube repositioning, allowing examination of task-consistent but context-varying control behaviors.

The observed activation trajectories show that module contributions fluctuate systematically with changes in target position and task phase, rather than remaining fixed or random. This behavior indicates that the learned representation dynamically reweights human ADL-derived sensorimotor primitives in response to environmental context, enabling adaptive feature recomposition without additional retraining. In particular, high activation shifts between modules correspond to transitions between spatial phases of the motion (e.g., approach, contact, stabilization), suggesting that each skill module encodes distinct yet complementary motor patterns that can be selectively reused. Such context-sensitive activation supports the framework's ability to generalize across new surgical conditions while maintaining coherent motor control—providing empirical evidence that the modular successor feature representation captures transferable, interpretable structure from human ADL data.

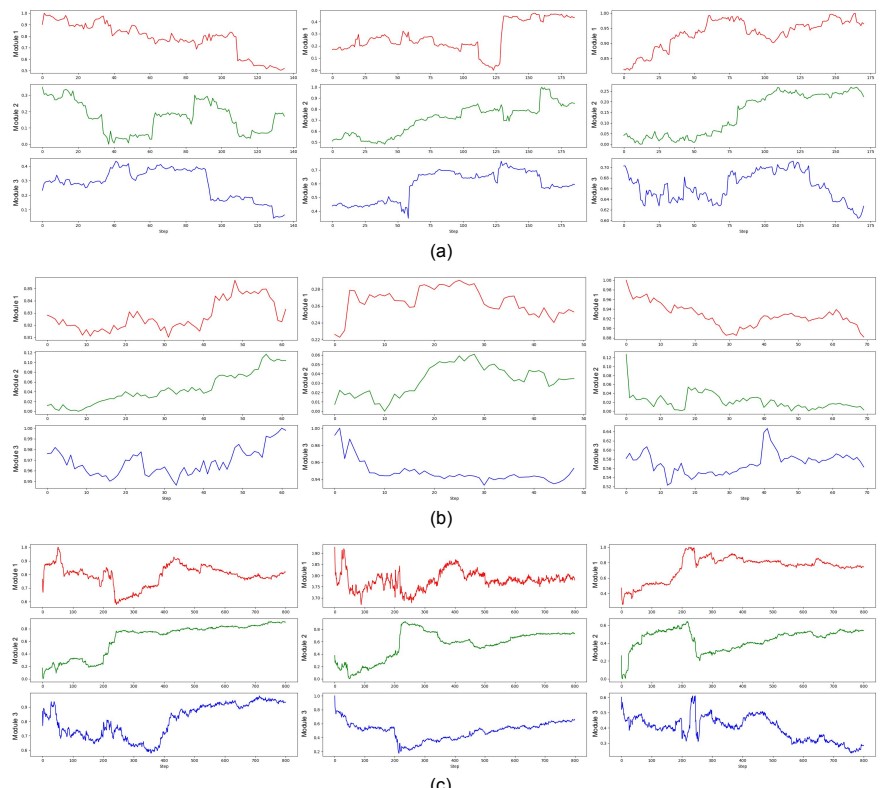

Figure 12: Temporal activation patterns of three ADL-derived skill modules during (a) PSM reaching gauze, (b) ECM localizing a static cube, and (c) ECM tracking a moving cube. Each row corresponds to a distinct skill module pretrained on *spoon–pick*, *cup–scoop/pour*, and *fork–spear* behaviors. The plotted values represent normalized module activations (i.e., each module's relative contribution to the current value estimate $\psi(s, a)$ over time). Variation across tasks and target instances shows that the framework dynamically reweights human-derived motor primitives based on task context, rather than relying on fixed feature combinations. This adaptive recomposition property enables generalization to new surgical conditions without retraining.

## 7.2   Effect of Skill Module Design

To assess the impact of modular representation design, we compare five configurations: (i) Module 1, (ii) Module 2, (iii) Module 3, (iv) a non-modular baseline trained jointly on all three DIM subsets without decomposition, and (v) our proposed compositional model integrating Modules 1–3. All models share identical network capacity and training budget to isolate the effect of modular structure. Figure 13 reports success rate as a function of interaction steps across three surgical tasks.

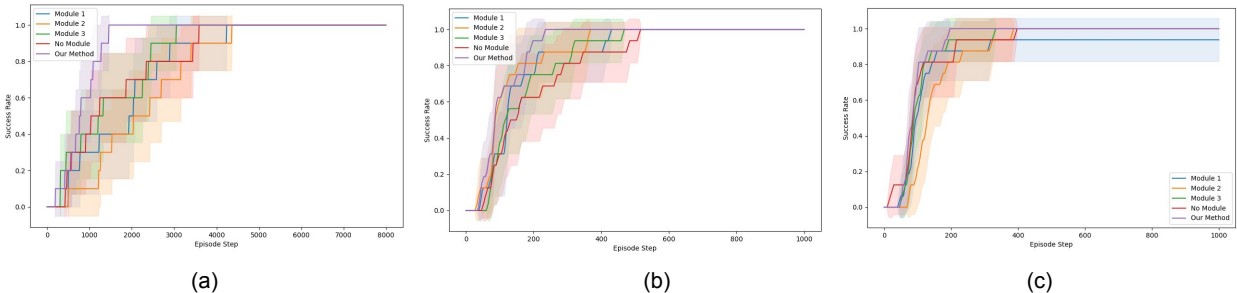

Figure 13: Success-rate learning curves comparing individual ADL modules (Modules 1–3), a non-modular baseline trained jointly on pooled DIM data, and our compositional model integrating Modules 1–2–3 across (a) PSM reaching gauze, (b) ECM localizing a static cube, and (c) ECM tracking a moving cube. All models are matched in network capacity and training budget to ensure a fair comparison. Single modules exhibit partial task specialization, whereas the compositional successor-feature representation achieves consistently faster convergence and higher stability across all tasks. These results demonstrate that the observed gains stem from structured module composition rather than model size. Compositional transfer works in surgical robot task learning!

Across all experiments, the compositional model consistently achieves faster convergence and higher stability than both the monolithic and single-module baselines. In the PSM reaching gauze task (Fig. 13a), all variants eventually attain near-perfect success, yet our composed model reaches this performance in substantially fewer steps, confirming that structured integration of ADL modules provides a stronger inductive bias for efficient policy learning. In the ECM static localization task (Fig. 13b), individual modules show similar early learning patterns, but their joint composition yields more stable convergence, indicating improved spatial goal representation through multi-skill fusion. In the ECM active tracking task (Fig. 13c), where dynamic adaptation is required, the composed model achieves the highest and most consistent success rate, demonstrating that recombining ADL-derived motor primitives enhances generalization under time-varying conditions. The results in Fig.13 show that the proposed modular design consistently outperforms this single-skill baseline, achieving at least 55.2%, 40.8%, and 41.4% improvements in sample efficiency on the PSM reaching task, ECM static localization task, and ECM dynamic tracking task, respectively (Hu et al., 2025). Overall, these results confirm that modular successor-feature composition—not increased network capacity—drives both the sample-efficiency gains and stability improvements observed in surgical task learning.

From these experiment results, we can find that using single/no module can also converge to a a success rate of 1. Task difficulty is one contributing factor. The evaluated surgical tasks are intentionally selected as foundational surgical skills rather than highly complex manipulation tasks, which makes them learnable even with limited inductive bias, provided sufficient interaction data. Moreover, in the single-module setting, each module is pretrained on a human ADL dataset that is semantically and kinematically related to the target task. Within the successor feature framework, this prior alignment enables the agent to eventually recover suitable task-specific reward weights through online adaptation. Consequently, given enough exploration, both single-module and no-module baselines can asymptotically achieve high success rates.

As shown in Fig. 13(c), Module 1 underperforms the no-module baseline. This behavior can be attributed to task structure and dataset coverage. The ECM dynamic tracking task fundamentally differs from the static localization tasks in Fig. 13(a,b). Dynamic tracking makes performance relative to the diversity and completeness of motion patterns available in the pretrained skill representations. The no-module baseline is trained on a mixture of all ADL datasets, which increases behavioral diversity and captures a broader range of motion primitives. This aggregated representation better spans the state–action space needed for dynamic tracking, thereby compensating for the deficiencies of any single skill source. Therefore, for this particular task, the mixed ADL dataset provides a more comprehensive and transferable motion representation than Module 1 alone, leading to superior performance despite the absence of modular decomposition.

## 7.3 Effect of ADL Pretraining Set Composition

To investigate how the composition of ADL pretraining modules influences transfer effectiveness, we examine the PSM gauze-reaching task using three distinct module sets, each representing a different subset of human daily activities: (1) **ADL Set 1**: *spoon–pick*, *fork–spear*, and *spoon–scoop/pour*; (2) **ADL Set 2**: *spoon–pick*, *fork–spear*, and *pizza–wheel*; and (3) **Ours**: *spoon–pick*, *fork–spear*, and *measuring–cup scoop/pour*. All models share identical architectures, data budgets, and optimization hyperparameters to isolate the effect of pretraining task selection.

As shown in Figure 14, *ADL Set 1* converges more quickly and attains higher asymptotic success than *ADL Set 2*, highlighting that the nature of the ADL pretraining tasks—not merely their number—determines transfer quality. Module compositions emphasizing smooth approach and controlled placement motions yield stronger inductive biases and more sample-efficient adaptation. Our chosen configuration (*Ours*) further improves both convergence speed and final performance, suggesting that the *measuring–cup scoop/pour* behavior provides motor primitives that align more closely with the fine, continuous motion patterns required in surgical manipulation.

It is noted that the observed performance drop occurs when the pretraining set includes a dissimilar human activity alongside relevant ones. While the attention mechanism reduces the contribution of less relevant modules, the presence of mismatched dynamics can still introduce representation noise, particularly when the total number of modules is small. When the skill pool contains predominantly relevant activities or is sufficiently large, attention robustly emphasizes useful modules and mitigates the impact of unrelated ones, as demonstrated in later ablations with randomly sampled larger module sets in section 7.4.

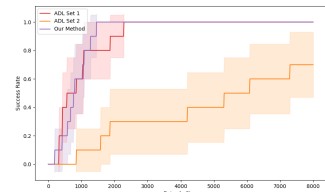

Figure 14: Effect of ADL pretraining set composition on the PSM reaching–gauze task. Curves compare two alternative module sets—ADL Set 1: *hand–wipe, stir–cup, place–object*, and ADL Set 2: *hammer, paint, sweep*—against our selected configuration: *spoon–pick, fork–spear, measuring–cup scoop/pour*. All variants use identical network architectures, parameter counts, and training budgets to ensure fair comparison.

To further interpret the differences observed across ADL module sets, Figure 15 visualizes representative 3D trajectories from the human ADL dataset, corresponding to the five tool-use behaviors considered in our pretraining pool.

Each plot depicts normalized end-effector motion traces extracted from human demonstrations, revealing distinct kinematic structures across activity types. Tasks such as *spoon–pick*, *fork–spear*, and *measuring–cup scoop/pour* exhibit smooth, continuous trajectories with stable tool–target contact phases and fine-grained control of motion curvature. In contrast, behaviors such as *hammer* or *pizza–wheel* involve abrupt, ballistic, or cyclic dynamics that differ substantially from the precision motions required in surgical manipulation.

These qualitative patterns explain the quantitative differences reported in Figure 14: pretraining modules that emphasize continuous approach and controlled placement behaviors yield transferable visuomotor representations more compatible with surgical tool dynamics. Thus, the diversity and kinematic similarity of ADL motion primitives jointly determine transfer efficacy, supporting our claim that appropriate ADL set design acts as a controllable prior over motor structure in surgical robot learning.

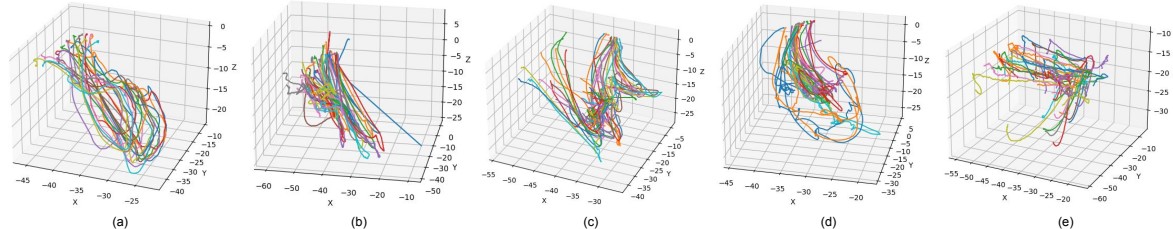

Figure 15: Visualization of human Activities of Daily Living (ADL) trajectories used for pretraining, derived from end-effector motion in 3D space. Shown are representative tool-use behaviors: (a) *spoon–pick*, (b) *fork–spear*, (c) *spoon–scoop/pour*, (d) *measuring–cup scoop/pour*, and (e) *pizza–wheel*. All trajectories are aligned to the tool tip coordinate frame and normalized for translation and scale to enable cross-task comparison. The plots illustrate complementary motion primitives—linear approach, rotational sweeping, and circular manipulation—that collectively provide diverse motor patterns for representation pretraining.

Overall, these findings reinforce that the ADL module set acts as a controllable prior over motor structure: coverage of complementary motion types (e.g., reach, grasp, and place) enhances representational richness and stabilizes transfer. This insight points toward future work on automatic ADL set selection or weighting mechanisms to optimize cross-domain transfer for specific surgical tasks.

## 7.4 Effect of the Number of Skill Modules

We study how the number of pretrained human skill modules affects learning efficiency and robustness in robotic surgical task learning. Specifically, we evaluate five configurations in which skill modules are randomly sampled from the DIM human activity dataset: (i) all 32 activities, (ii) 25 activities, (iii) 20 activities, (iv) 15 activities, and (v) 10 activities. Across all settings, the network architecture, total parameter count, and training budget are kept identical, ensuring that any observed performance differences can be attributed solely to the number of available skill modules rather than model capacity.

Figure 16 reports success-rate learning curves as a function of interaction steps for three representative surgical tasks. In the PSM reaching gauze task (Fig. 16a), increasing the number of skill modules consistently accelerates learning, indicating that a larger pool of human motor primitives provides richer transferable structure for fine-grained manipulation. For ECM localizing a static cube and ECM tracking a moving cube (Fig. 16b,c), performance generally improves as more modules are introduced, but becomes less sensitive beyond a moderate scale: using 32 modules achieves performance comparable to using 25 modules. Notably, in both ECM tasks, as few as 10 randomly selected skill modules attain performance close to that obtained with 20 modules. These results demonstrate that the proposed framework does not rely on carefully curated human activity sets and can effectively leverage a small, randomly sampled subset of skills for robust surgical task learning.

To further elucidate this behavior, Figure 17 visualizes the temporal activation patterns of all 32 skill modules during task execution. Each row corresponds to a skill module pretrained on a distinct human activity, and color intensity denotes its normalized contribution to the value estimate $\psi(s, a)$ over time. The resulting patterns show that different surgical tasks selectively engage different subsets of modules and dynamically adjust their relative contributions. For the PSM reaching gauze and ECM static localization tasks, module activations vary substantially over time, reflecting transitions across task phases and changing coordination requirements. In contrast, during ECM tracking of a moving cube, activation patterns remain comparatively stable, consistent with the sustained and continuous control demands of visual tracking.

## 7.5 Effect of Dataset Source

To investigate how the choice of human activity dataset influences transfer effectiveness in robotic surgical task learning, we compare two representative ADL sources: the EgoDex dataset and the DIM dataset. The DIM dataset contains 1,603 motion trials across 32 daily manipulation tasks, such as unlocking a lock with a key and flipping bread. EgoDex contains approximately 829 hours of egocentric video with 3D hand finger pose across 194 daily manipulation tasks involving everyday objects, like chessboard and tupperware (Hoque

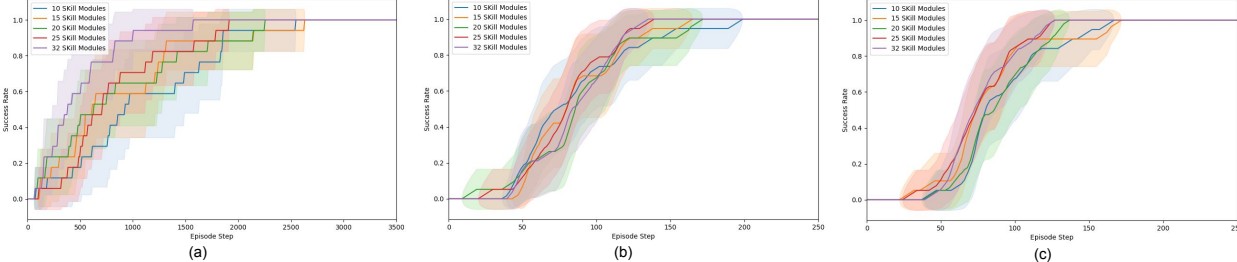

Figure 16: Success-rate learning curves with different numbers of pretrained human skill modules randomly sampled from the DIM activity set ($N \in 10, 15, 20, 25, 32$). (a) PSM reaching gauze, (b) ECM localizing a static cube, and (c) ECM tracking a moving cube. Curves report mean success rate over training steps, with shaded regions indicating variability across runs. Increasing the number of available modules consistently accelerates learning in the PSM reaching task. For both ECM tasks, performance improves with additional modules but becomes less sensitive beyond a moderate scale, with comparable results achieved using 25 and 32 modules. Notably, even a small randomly sampled subset of modules (e.g., $N = 10$) attains performance close to larger module sets, demonstrating the robustness of the proposed framework to module selection.

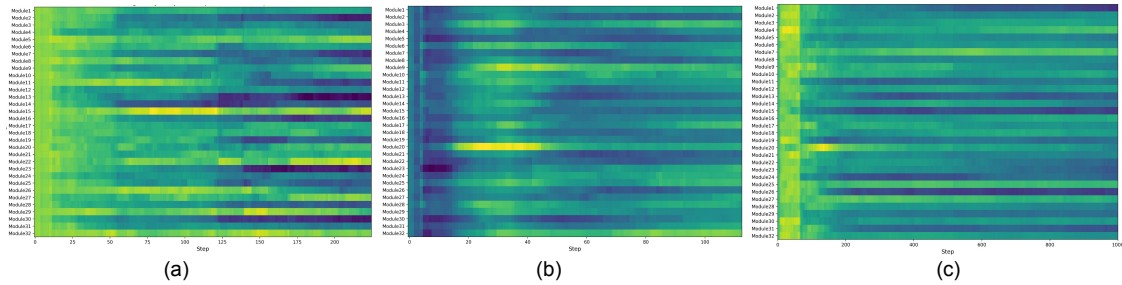

Figure 17: Temporal activation patterns of 32 pretrained skill modules across three surgical tasks: (a) PSM reaching gauze, (b) ECM localizing a static cube, and (c) ECM tracking a moving cube. Each row corresponds to a skill module pretrained on a single human activity from the DIM dataset. Color intensity denotes the normalized activation of each module, representing its relative contribution to the value estimate $\psi(s, a)$ over time. Distinct surgical tasks induce different activation profiles across modules. For PSM reaching gauze and ECM static localization, module activations vary over time, reflecting task-phase–dependent coordination. In contrast, ECM tracking exhibits relatively stable activation patterns, indicating consistent module engagement required for continuous tracking behavior.

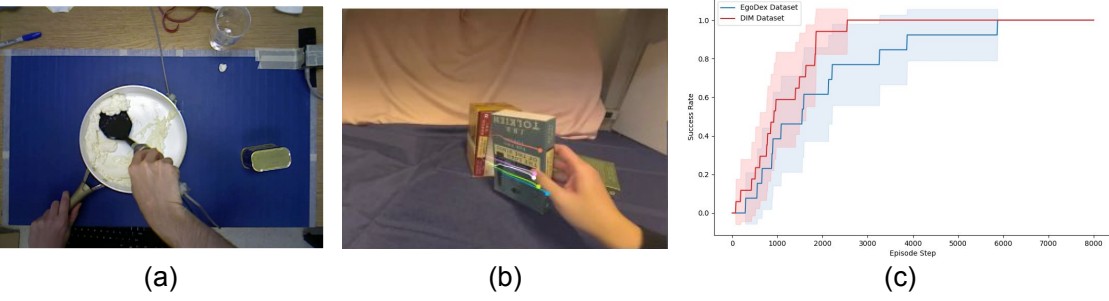

(a)  (b)  (c)

Figure 18: Effect of ADL dataset source on the PSM reaching–gauze task. (a) Representative human activities from the DIM dataset, characterized by tool-mediated manipulation. (b) Representative human activities from the EgoDex dataset, emphasizing bare-hand motion. (c) Learning curves comparing transfer performance when pretraining on DIM versus EgoDex. Pretraining with DIM consistently achieves higher success, highlighting the advantage of tool-centered ADL data due to its closer structural alignment with surgical instrument control.

et al., 2025). The dataset provides detailed 3D position and orientation of all upper body joints (including 25 joints for each hand, e.g., leftIndexFingerIntermediateBase, leftIndexFingerIntermediateTip, and leftIndexFingerKnuckle). In our experiment, only the 3D positions of the right thumb tip from EgoDex are used as the transferred human motion level information.

To ensure a fair comparison between datasets, we select an equal number of human activities from each source. Specifically, we randomly sample 15 activities from the 32 available tasks in the DIM dataset. For the EgoDEX dataset, we construct a comparable 32-activity subset that closely matches the types of daily manipulation tasks in DIM, differing primarily in that EgoDEX provides hand-centered trajectories whereas DIM provides tool-centered trajectories. Then we randomly sample 15 activities from this matched subset. The experimental results are summarized in Fig. 18.

The observed performance gap can be explained by the fundamental difference in motion structure between tool-centered and hand-centered data. The DIM dataset records tool-centered motion trajectories during human daily manipulation, while the EgoDEX dataset captures hand-centered trajectories when humans perform daily tasks with or without tools. In certain activities, such as cutting or stirring, hand and tool motions may be closely aligned, since the hand directly guides the tool along a similar path. However, in many manipulation tasks, the kinematic patterns of the hand and the effective tool motion differ substantially. For example, during screw driving, the human hand primarily exhibits oscillatory wrist rotations, whereas the trajectory of the screwdriver tip follows a smooth and continuous circular motion that directly reflects the task-relevant interaction with the object. This tool-mediated trajectory more accurately represents the functional end-effector motion. Compared to hand-centered data, tool-centered interaction trajectories therefore better align with the control paradigm of surgical robotic tasks, which similarly rely on precise instrument motion rather than articulated finger movements. As a result, tool-centered ADL datasets provide more directly transferable motor-level structure for learning surgical manipulation skills.

Overall, these results indicate that ADL datasets emphasizing tool-centered interaction provide more effective inductive biases for robotic surgical task learning than datasets centered on bare-hand motion. This finding supports the central hypothesis of our work: surgical expertise is not an isolated or domain-specific capability, but rather an extension of everyday tool-use behaviors that are progressively refined and specialized through practice.

### 7.6 Experiments Discussion and Implications

**Pre-training Data.** All pre-training data are human-hand demonstrations from the DIM dataset, without relying on tele-operated robotic data. Our assumption is that when humans manipulate everyday tools, their hand motions exhibit underlying motor-level regularities—such as approach, alignment, and constrained

movement—that are also required in advanced surgical manipulation. The observed cross-embodiment generalization originates from motor-level representation abstraction, not from exposure to multiple embodiments during pre-training.

**Representation Capacity.** The results above demonstrate that modular successor-feature representations trained on human demonstrations capture transferable structure in both spatial and visuomotor dynamics. This supports the theoretical premise that nonlinear task rewards can be linearly composed in a shared predictive space, allowing adaptive recombination without retraining.

**Evaluation Design.** The selected surgical subtasks—precise positioning and visual tracking—reflect foundational motor and perceptual skills shared with human ADL behaviors. These settings were intentionally chosen to isolate cross-domain transfer effects, as our human dataset does not include grasping or bimanual coordination. Thus, the evaluation focuses on verifying skill transfer under well-aligned kinematic and perceptual structures rather than sim-to-real adaptation.

**Computational and Runtime Efficiency.** The proposed transfer mechanism operates in real time at 10 Hz on an RTX 3060 GPU. Since inference involves a lightweight inner product between successor features and the estimated reward weights, policy adaptation requires no gradient updates or rollouts. This property is particularly valuable for low-latency surgical applications.

**Relation to Imitation Learning.** Unlike conventional imitation learning—which depends on task-specific demonstrations—our approach distills general behavioral priors from diverse human activities and reuses them through compositional reasoning. This decoupling of representation learning from task execution offers a scalable path toward data-efficient, safety-aware surgical policy learning.

**Broader Implications.** Overall, these findings underscore the potential of modular, human-derived feature spaces as reusable priors for embodied surgical robots. Future work will explore automated module selection and integration with real-time visual feedback loops for closed-loop autonomy.

**Broader Extensions.** Regarding extension to contact-rich manipulation, our approach can augment the state with contact-relevant variables, such as force/torque signals, gripper indicators (close/open), or constraint-related features. Our framework can be learned over this augmented state, enabling the model to capture contact-conditioned transition dynamics—that is, how both motion and contact states evolve over time under different actions.

## 8 Conclusions and Limitations

This work presents a principled framework for *cross-domain surgical robot task learning* through transferable motor representations learned from human Activities of Daily Living (ADL). By introducing a *Predictive State–Action Representation Space* constructed via modular successor features, we enable policy transfer across embodiment and task boundaries. The proposed approach captures shared motor dynamics and variability inherent in everyday actions, allowing robotic surgical systems to acquire new skills with minimal in-domain data. Empirical results on both simulated and real-world da Vinci platforms confirm that our method achieves rapid convergence, high sample efficiency, and robust adaptation across tool and camera control tasks—highlighting its potential as a scalable foundation for data-efficient surgical robot learning.

While promising, this framework also exposes several opportunities for future research. First, the current formulation emphasizes motion-level state–action representation and does not yet explicitly encode gripper interactions critical for fine manipulation such as suturing or tissue handling. Extending the representation to include contact and force modalities would strengthen its expressiveness for bimanual and deformable-object tasks. Second, although our modular successor features capture visual and kinematic variability, incorporating multimodal sensory feedback (e.g., force, tactile, and depth cues) would enhance safety and precision under clinical uncertainty. Finally, the present experiments focus on low-level subskills—reaching, localization, and tracking—chosen for their controllable correspondence with ADL motions. Future work will explore hierarchical composition of modules for more complex surgical workflows, integrating semantic and task-level reasoning within the same predictive representation space.

Overall, our study provides evidence that human motor priors distilled from ADL can serve as a transferable substrate for robotic surgical task learning, bridging the gap between general human dexterity and domain-specific surgical autonomy.

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

# A    Algorithm Pseudocode

---

**Algorithm 1:** Cross-Domain Surgical Skill Transfer via Modular Successor Features and GPI

---

**Require:** ADL dataset $\mathcal{D}_{\text{ADL}} = \bigcup_{k=1}^{n} \mathcal{D}_{\text{ADL}}^{(k)}$ with $n$ skill modules

1: Optional: small dataset of surgical interactions $\mathcal{D}_{\text{Surg}} = \{(s, a, r)\}$

2:

**Ensure:** Adapted surgical policy $\pi$

3: **Stage 1: Modular Representation Learning from ADL for** $k = 1$ *to* $n$ **do**

4:

      **end**

      Initialize module parameters $\theta^{(k)}$ for cumulants $\phi^k$ and SFs $\psi^k$ **for** *each* $(s_t, a_t, r_t, s_{t+1})$ *in* $\mathcal{D}_{ADL}^k$

    **do**

5:

      **end**

      Update $\phi^k(s_t, a_t, s_{t+1}) \leftarrow$ learned cumulant dynamics

6: Encode the expected discounted future cumulants to get the successor features

$$\psi^k(s_t^k, a_t) = \mathbb{E}_\pi \left[ \sum_{i=0}^{\infty} \gamma^i \phi_{t+i}^k \right]$$

7: Construct joint representations by concatenating module-level cumulants:

$$\phi_t = \left[ \phi_t^1, \ldots, \phi_t^n \right], \quad \psi_t = \left[ \psi_t^1, \ldots, \psi_t^n \right].$$

8: Update $\psi$ and $w$ through loss:

$$\mathcal{L}_\psi \leftarrow \| \phi_t + \gamma \psi_{t+1} - \psi_t \|^2$$
$$\mathcal{L}_\phi \leftarrow \left\| r_t - \phi_t^\top w_t \right\|^2$$

9:

10:

11: **Stage 2: Reward Adaptation for Surgical Task**

12: Compute modular reward weights $w_{\text{Surg}}^*$:

$$w_{\text{Surg}}^* = \arg \min_w \mathbb{E}_{(s,a,r) \sim \mathcal{D}_{\text{Surg}}} \left[ \left( r_{\text{Surg}} - \sum_{k=1}^{K} \phi^k(s, a) w^k \right)^2 \right]$$

13: **Stage 3: Policy Synthesis via Generalized Policy Improvement (GPI) for** *each decision step* **do**

14:

      **end**

      Observe current state $s$

15: Compute:

$$a = \arg \max_{a \in \mathcal{A}_{\text{Surg}}} \max_{z \in \mathbb{M}_{\text{ADL}}} \left\{ \psi_\theta(s, a, z)^\top w_{\text{Surg}} \right\}$$

16: Execute action $a$, observe next state and reward

17:

18: **return** Adapted surgical policy $\pi(s) = a$

---

# B  Theoretical Results

## B.1  Proof of Theorem 1

**Proof:** Define the true Q-function and the approximated Q-function:

$$Q_\pi(s,a) = \mathbb{E}\left[\sum_{t=0}^{\infty} \gamma^t r_{\text{Surg}}(s,a) \,\Big|\, s_0 = s,\, a_0 = a\right],$$

$$\hat{Q}_\pi(s,a;\hat{w}) = \mathbb{E}\left[\sum_{i=0}^{\infty} \gamma^t \hat{r}_{\text{Surg}}(s,a;\hat{w}) \,\Big|\, s_0 = s,\, a_0 = a\right].$$

By the Bellman equation, for any $(s,a)$:

$$Q_\pi(s,a) = \int r\, dR_{s,a}(r) + \gamma \sum_{s',a'} p(s'|s,a)\pi(a'|s')Q_\pi(s',a'),$$

$$\hat{Q}_\pi(s,a;\hat{w}) = \hat{r}_{\text{Surg}}(s,a;\hat{w}) + \gamma \sum_{s',a'} p(s'|s,a)\pi(a'|s')\hat{Q}_\pi(s',a';\hat{w}).$$

Thus, the error is:

$$\left|Q_\pi(s,a) - \hat{Q}_\pi(s,a;\hat{w})\right| = \left|\int r\, dR_{s,a}(r) - \hat{r}_{\text{Surg}}(s,a;\hat{w}) + \gamma \sum_{s',a'} p(s'|s,a)\pi(a'|s')\big(Q_\pi(s',a') - \hat{Q}_\pi(s',a';\hat{w})\big)\right|$$

$$\leq \left|\int r\, dR_{s,a}(r) - \hat{r}_{\text{Surg}}(s,a;\hat{w})\right| + \gamma \left|\sum_{s',a'} p(s'|s,a)\pi(a'|s')\big(Q_\pi(s',a') - \hat{Q}_\pi(s',a';\hat{w})\big)\right|.$$

Since $\sum_{s',a'} p(s'|s,a)\pi(a'|s') = 1$, we have:

$$\left|Q_\pi(s,a) - \hat{Q}_\pi(s,a;\hat{w})\right| \leq \left|\int r\, dR_{s,a}(r) - \hat{r}_{\text{Surg}}(s,a;\hat{w})\right| + \gamma \|Q_\pi - \hat{Q}_\pi(\hat{w})\|_\infty.$$

Taking the supremum over $s,a$:

$$\|Q_\pi - \hat{Q}_\pi(\hat{w})\|_\infty \leq \sup_{s,a} \left|\int r\, dR_{s,a}(r) - \hat{r}_{\text{Surg}}(s,a;\hat{w})\right| + \gamma \|Q_\pi - \hat{Q}_\pi(\hat{w})\|_\infty.$$

Rearranging terms:

$$(1-\gamma)\|Q_\pi - \hat{Q}_\pi(\hat{w})\|_\infty \leq \sup_{s,a} \left|\int r\, dR_{s,a}(r) - \hat{r}_{\text{Surg}}(s,a;\hat{w})\right|,$$

$$\|Q_\pi - \hat{Q}_\pi(\hat{w})\|_\infty \leq \frac{1}{1-\gamma} \sup_{s,a} \left|\int r\, dR_{s,a}(r) - \hat{r}_{\text{Surg}}(s,a;\hat{w})\right|.$$

Now, bound the right-hand side term. Since the state-action space is finite:

$$\sup_{s,a} \left|\int r\, dR_{s,a}(r) - \hat{r}_{\text{Surg}}(s,a;\hat{w})\right| \leq \sqrt{\sum_{s,a} \left(\int r\, dR_{s,a}(r) - \hat{r}_{\text{Surg}}(s,a;\hat{w})\right)^2}.$$

Using $\rho(s,a) \geq \rho_{\min} > 0$:

$$\sum_{s,a}(\int r\, dR_{s,a}(r) - \hat{r}_{\text{target}}(s,a;\hat{w}))^2 \leq \frac{1}{\rho_{\min}} \sum_{s,a} \rho(s,a) \left(\int r\, dR_{s,a}(r) - \hat{r}_{\text{Surg}}(s,a;\hat{w})\right)^2$$

$$= \frac{1}{\rho_{\min}} \mathbb{E}_{(s,a)\sim\rho}\left[\left(\int r\, dR_{s,a}(r) - \hat{r}_{\text{Surg}}(s,a;\hat{w})\right)^2\right].$$

Therefore:

$$\sup_{s,a}\left|\int r\, dR_{s,a}(r) - \hat{r}_{\text{Surg}}(s,a;\hat{w})\right| \leq \frac{1}{\sqrt{\rho_{\min}}} \sqrt{\mathbb{E}_{(s,a)\sim\rho}\left[\left(\int r\, dR_{s,a}(r) - \hat{r}_{\text{Surg}}(s,a;\hat{w})\right)^2\right]}.$$

For each $(s,a)$, by Jensen's inequality:

$$\left(\int r\, dR_{s,a}(r) - \hat{r}_{\text{Surg}}(s,a;\hat{w})\right)^2 = \left(\mathbb{E}_{r\sim R_{s,a}}[r] - \hat{r}_{\text{Surg}}(s,a;\hat{w})\right)^2 \leq \mathbb{E}_{r\sim R_{s,a}}\left[(r - \hat{r}_{\text{Surg}}(s,a;\hat{w}))^2\right].$$

Hence:

$$\mathbb{E}_{(s,a)\sim\rho}\left[\left(\int r\, dR_{s,a}(r) - \hat{r}_{\text{Surg}}(s,a;\hat{w})\right)^2\right] \leq \mathbb{E}_{(s,a)\sim\rho}\mathbb{E}_{r\sim R_{s,a}}\left[(r - \hat{r}_{\text{Surg}}(s,a;\hat{w}))^2\right].$$

That is:

$$||Q_\pi - \hat{Q}_\pi(\hat{w})||_\infty \leq \frac{1}{1-\gamma} \cdot \frac{1}{\sqrt{\rho_{\min}}} \sqrt{\mathbb{E}_{(s,a)\sim\rho,\, r\sim R_{s,a}}[(r - \hat{r}_{\text{target}}(s,a;\hat{w}))^2]}.$$

By standard uniform convergence results for linear regression with bounded features, it has $1-\delta$ posibility that:

$$\mathbb{E}_{(s,a)\sim\rho,\, r\sim R_{s,a}}\left[(r - \hat{r}_{\text{Surg}}(s,a;\hat{w}))^2\right] \leq \inf_w \mathbb{E}\left[\left(r - \phi(s,a)^\top w\right)^2\right] + C\,(BR + BW)^2 \sqrt{\frac{d + \log(1/\delta)}{n}}.$$

It yields the final result:

$$||Q_\pi - \hat{Q}_\pi(\hat{w})||_\infty \leq \frac{1}{(1-\gamma)\sqrt{\rho_{\min}}} \sqrt{\inf_w \mathbb{E}\left[\left(r_{\text{Sur}}(s,a) - \phi(s,a)^\top w\right)^2\right] + C\,(BR + BW)^2 \sqrt{\frac{d + \log(1/\delta)}{n}}}.$$

∎

## B.2 Proof of Theorem 2

**Proof:** From Theorem 1, we have for any policy $\pi$:

$$||Q_\pi(s,a) - \hat{Q}_\pi(s,a;\hat{w})||_\infty \leq \frac{1}{(1-\gamma)\sqrt{\rho_{\min}}} \sqrt{\inf_w \mathbb{E}\left[\left(r_{\text{Surg}}(s,a) - \phi(s,a)^\top w\right)^2\right] + C\,(BR + BW)^2 \sqrt{\frac{d + \log(1/\delta)}{n}}}.$$

Denote

$$Q^{\max}(s,a) := \max_{\pi\in\Pi} Q_\pi(s,a), \quad \hat{Q}^{\max}(s,a) := \max_{\pi\in\Pi} \hat{Q}_\pi(s,a).$$

Since the maximum absolute difference is bounded by the supremum norm, we have:

$$\left|Q^{\max}(s,a) - \hat{Q}^{\max}(s,a)\right| \leq C.$$

Define the Bellman operator $\mathcal{T}^\pi$ for policy $\pi$ as:

$$\mathcal{T}^\pi(Q^\pi(s,a)) = r(s,a,s') + \gamma Q^\pi(s',\pi(s')).$$

Then, for all $s,a \in \mathcal{S} \times \mathcal{A}$ and $j = 1,\ldots,M$:

$$\begin{aligned}
\mathcal{T}^{\pi'}(\hat{Q}^{\max}(s,a)) &= r(s,a,s') + \gamma \hat{Q}^{\max}(s',\pi'(s')) \\
&\geq r(s,a,s') + \gamma Q^{\max}(s',\pi'(s')) - \gamma C \\
&\geq r(s,a,s') + \gamma Q^\pi(s',\pi_j(s')) - \gamma C \\
&= \mathcal{T}^\pi(Q^\pi(s,a)) - \gamma C \\
&= Q^\pi(s,a) - \gamma C.
\end{aligned}$$

Since this inequality holds for all policies $\pi$, we have:

$$\begin{aligned}
\mathcal{T}^{\pi'}(\hat{Q}^{\max}(s,a)) &\geq \max_{\pi \in \Pi} Q_\pi(s,a) - \gamma C \\
&= Q^{\max}(s,a) - \gamma C \\
&\geq \hat{Q}^{\max}(s,a) - (1+\gamma)C.
\end{aligned}$$

Applying the operator $\mathcal{T}^{\pi'}$ repeatedly:

$$\begin{aligned}
(\mathcal{T}^{\pi'})^2(\hat{Q}^{\max}(s,a)) &\geq \mathcal{T}^{\pi'}(\hat{Q}^{\max}(s,a)) - \gamma(1+\gamma)C \\
&\geq \hat{Q}^{\max}(s,a) - (1+\gamma)C - \gamma(1+\gamma)C.
\end{aligned}$$

By induction, after $n$ iterations:

$$(\mathcal{T}^{\pi'})^n(\hat{Q}^{\max}(s,a)) \geq \hat{Q}^{\max}(s,a) - (1+\gamma)C \sum_{k=0}^{n-1} \gamma^k.$$

Taking the limit as $n \to \infty$:

$$Q_{\pi'}(s,a) = \lim_{n\to\infty} (\mathcal{T}^{\pi'})^n(\hat{Q}^{\max}(s,a)) \geq \hat{Q}^{\max}(s,a) - \frac{1+\gamma}{1-\gamma}C.$$

Finally, using the bound on $\hat{Q}^{\max}$:

$$Q_{\pi'}(s,a) \geq \max_{\pi \in \Pi} Q_\pi(s,a) - \frac{2}{(1-\gamma)^2\sqrt{\rho_{\min}}}\sqrt{\inf_w \mathbb{E}\left[\left(r_{\text{Surg}}(s,a) - \phi(s,a)^\top w\right)^2\right] + C(BR+BW)^2 \sqrt{\frac{d+\log(1/\delta)}{n}}}.$$

∎

## C  Experiment Setup

### C.1  Human ADL Dataset

For this study, we utilized the DIM dataset, a comprehensive collection of human demonstrations on task-oriented interactive manipulation in daily scenarios. The DIM dataset consists of two major parts: (1) a diverse set of 1,603 trials covering 32 types of daily motions, and (2) a specialized subset of 1,751 trials focused exclusively on the pouring motion, collected under varying environmental conditions. For the purpose of our

research, we primarily focus on the Cartesian position data, as it provides a domain-agnostic representation of object-centric task dynamics that is directly relevant to robotic manipulation and facilitates the construction of transferable state-action representations.

In our work, we model each human ADL skill as a skill module. However, due to the limited amount of data available in each human daily activity of the DIM dataset, training robust and generalizable skill modules directly from the raw data is challenging. To address this limitation, we apply data augmentation techniques by translating and rotating the demonstration trajectories, thereby enriching the diversity of the training set and improving the robustness of the learned skill modules.

## C.2 Simulation Experiment

**PSM Reaching Gauze.** The PSM is a critical component of robot-assisted surgery, enabling precise tool positioning relative to tissue or surgical platforms. In this task, the objective is to position the PSM jaw tip slightly above a gauze pad, which is randomly placed on a surgical tray at the start of each episode. The jaw remains in a fixed orientation during the task. The observation space is defined as the 3D Cartesian coordinates of the end-effector, denoted as $\mathbf{s}_t \in \mathbb{R}^3$, where $\mathbf{s}_t = (x_t, y_t, z_t)$. The action space is defined as the continuous Cartesian displacement of the end-effector, denoted as $\mathbf{a}_t \in \mathbb{R}^3$. The reward at each timestep is defined as $r_t = -\|\mathbf{s}_t - \mathbf{g}\|_2$, where $\mathbf{g} \in \mathbb{R}^3$ is the position of the gauze and $\|\cdot\|_2$ denotes the Euclidean norm. A trial is considered successful if the distance satisfies $\|\mathbf{s}_t - \mathbf{g}\|_2 < 5\,\text{mm}$. Each episode is limited to 300 timesteps.

**ECM Localizing Static Cube.** The ECM provides a stable visual perspective of the surgical site. This task focuses on localizing a static target cube at the center of the ECM camera view. The observation space consists of three consecutive frames, $s_{t-2}$, $s_{t-1}$, and $s_t$, where each $s_t \in \mathbb{R}^2$ represents the pixel centroid of the target in the camera image. The continuous action space controls the first two ECM joint positions, denoted as $\mathbf{a}_t \in \mathbb{R}^2$. The reward at each timestep is defined as $r_t = -\|s_t - c\|_2$, where $c \in \mathbb{R}^2$ is the pixel coordinate of the image center. At the beginning of each episode, the cube is placed randomly within the workspace. A trial is considered successful if the distance satisfies $\|s_t - c\|_2 < 10$ pixels. Each episode is limited to 150 timesteps.

**ECM Tracking Moving Cube.** Unlike the static localization task, this setting involves tracking a target cube that moves continuously within the camera's field of view. The goal is to maintain the cube near the center of the ECM image as it follows a random trajectory in the XY plane at a constant Z height. At each environment timestep, the cube advances twice along this path to simulate smooth and realistic motion, with updates handled by the underlying physics engine. The agent observes three consecutive target centroids in pixel space, $s_{t-2}, s_{t-1}, s_t \in \mathbb{R}^2$, capturing temporal movement patterns. Control is achieved by issuing continuous commands to the first two ECM joints, $\mathbf{a}_t \in \mathbb{R}^2$. The reward is computed as $r_t = -\|s_t - c\|_2$, where $c$ is the image center. A timestep is deemed successful if the tracking error is below 25 pixels. Each episode spans 800 steps, and success is evaluated by the proportion of timesteps that satisfy the tracking condition.

## C.3 Real-world Experiment

**PSM Reaching Gauze.** This real-world PSM task extends the simulated PSM reaching setup by incorporating a grasping action once the target is reached. The objective is to position the PSM jaw tip slightly above the gauze pad, and then perform a grasping maneuver. The observation space and action space are identical to the simulated task: the state is defined as the 3D Cartesian coordinates of the end-effector, $\mathbf{s}_t \in \mathbb{R}^3$, and the action space is the continuous Cartesian displacement, $\mathbf{a}_t \in \mathbb{R}^3$. When the PSM tip enters a predefined target area suitable for grasping, the gripper is automatically commanded to open and then close to grasp the gauze. This additional manipulation step introduces a combined reaching and grasping challenge, making the task closer to real surgical execution. In the experiment, eight gauze pads are randomly placed, and the PSM is allowed six attempts per gauze. Each attempt is constrained to a maximum of 200 timesteps.

**ECM Localizing Gauze.** This real-world task mirrors the simulated ECM localizing static cube task but focuses on localizing a gauze pad within the ECM camera view. The observation space, state, and action space are identical to the simulated setting: the state consists of three consecutive frames, $s_{t-2}, s_{t-1}, s_t \in \mathbb{R}^2$, representing the pixel centroid of the gauze, and the continuous action space controls the first two ECM joint positions, $\mathbf{a}_t \in \mathbb{R}^2$. In the real-world experiment, the centroid of the gauze in the ECM image view is obtained through image processing techniques. A trial is considered successful if the pixel distance satisfies $\|s_t - c\|_2 \leq 25$. This task setup reflects practical demands in real surgical scenarios, where the endoscopic camera must accurately localize and maintain the view of soft tissues or surgical materials to support downstream manipulation and ensure procedural safety.

**ECM Tracking Moving Tool.** This real-world task builds upon the simulated ECM tracking moving cube task by focusing on tracking a surgical tool operated by a human. A label is attached to the tool tip, allowing its pixel position to be extracted through image processing. The observation space, state, and action space remain consistent with the simulated setting: the state consists of three consecutive tool tip centroids, $s_{t-2}, s_{t-1}, s_t \in \mathbb{R}^2$, and the continuous action space controls the first two ECM joint positions, $\mathbf{a}_t \in \mathbb{R}^2$. During the experiment, a human operator holds the labeled tool and simulates surgical execution by continuously moving it to evaluate the ECM's active tracking performance. This setup closely reflects real surgical scenarios, where the endoscopic camera must dynamically follow moving instruments to maintain a stable and informative view of the surgical site.

## D   Implementation Details

### D.1   Transferable State-Action Representation Space

A fundamental challenge in leveraging human ADL datasets for robotic surgical task learning lies in the inherent mismatch between the two domains. The workspace of human ADL tasks is typically characterized by a much larger and more diverse range of motions, environments, and action scales compared to the constrained and highly specialized workspace of surgical robots. This discrepancy leads to a significant domain gap in both the state and action spaces, which, if left unaddressed, impedes effective knowledge transfer and degrades policy performance when transitioning from human demonstrations to surgical tasks.

To address this challenge, we introduce the concept of a *transferable state-action representation space*. The goal is to learn a shared space $\mathcal{Z}$, where both human and robot state-action pairs can be projected in a way that preserves task-relevant structure and dynamics. Concretely, we learn a mapping function $f : \mathcal{S}_H \times \mathcal{A}_H \to \mathcal{Z}$ that transforms human demonstrations into this shared space, as well as a complementary mapping $g : \mathcal{Z} \to \mathcal{A}_R$ that maps predicted actions from the shared space into robot-executable actions during downstream surgical tasks.

Before modeling the motion dynamics using the deep successor feature framework, we first apply the mapping $f$ to project the human ADL state-action pairs into the shared representation space $\mathcal{Z}$. This preprocessing step is crucial because it allows the model to operate over a common input domain, effectively mitigating the large domain gap between human and robotic data. Once the state-action data are embedded in $\mathcal{Z}$, we can then learn the successor features $\psi(z)$ over this space, which capture the expected discounted future feature occupancy under a given policy. By operating within this unified space, the successor feature model can learn transferable task dynamics that generalize across domains, facilitating effective policy learning and adaptation. During downstream deployment, the predicted actions in $\mathcal{Z}$ are decoded into the robot action space via the mapping $g$, ensuring that the learned policies are directly executable by the surgical robot.

### D.2   Training Details

For each human ADL dataset, we train a pair of models, $\phi$ and $\psi$, using offline trajectories. Specifically, we define $K$ neural networks $\phi_k(\cdot, \cdot; \theta_k^\phi) : \mathcal{S} \times \mathcal{A} \to \mathbb{R}^d$ and $K$ neural networks $\psi_k(\cdot, \cdot; \theta_k^\psi) : \mathcal{S} \times \mathcal{A} \to \mathbb{R}$, where each $\phi_k$ and $\psi_k$ pair is trained on one of the $K$ human ADL datasets. Here, $\theta_k^\phi$ and $\theta_k^\psi$ denote the learned parameters for dataset $k \in [K]$. Together, these $K$ pairs of networks generate feature mappings $[\phi_k(s, a)]_{k \in [K]} \in \mathbb{R}^{K \times d}$ and Q-value estimates $[\psi_k(s, a)]_{k \in [K]} \in \mathbb{R}^K$ for any state-action pair $(s, a)$, capturing

Table 3: Model hyperparameters across tasks.

| | PSM Reach | ECM Static | ECM Active |
|---|---|---|---|
| **COMBO** | | | |
| State dim | 3 | 2 | 6 |
| Action dim | 3 | 2 | 2 |
| Batch size | 128 | 128 | 128 |
| Discount | 0.99 | 0.99 | 0.99 |
| Target update | 0.005 | 0.005 | 0.005 |
| Entropy | auto | auto | auto |
| Policy hidden sizes | [256, 256] | [256, 256] | [256, 256] |
| $\log \sigma$ clamp | $[-20, 2]$ | $[-20, 2]$ | $[-20, 2]$ |
| Policy lr | $1 \times 10^{-5}$ | $1 \times 10^{-5}$ | $1 \times 10^{-5}$ |
| Q-net hidden sizes | [256, 256] | [256, 256] | [256, 256] |
| Q-net lr | $3 \times 10^{-4}$ | $3 \times 10^{-4}$ | $3 \times 10^{-4}$ |
| Backbone hidden sizes | [200, 200, 200, 200] | [200, 200, 200, 200] | [200, 200, 200, 200] |
| **MBPO** | | | |
| State dim | 3 | 2 | 6 |
| Action dim | 3 | 2 | 2 |
| Batch size | 128 | 128 | 128 |
| Discount | 0.99 | 0.99 | 0.99 |
| Target update | 0.005 | 0.005 | 0.005 |
| Clip grad norm | 10.0 | 10.0 | 10.0 |
| Actor hidden sizes | [256, 256] | [256, 256] | [256, 256] |
| Critic hidden sizes | [256, 256] | [256, 256] | [256, 256] |
| Dynamics ensemble size | 7 | 7 | 7 |
| Dynamics hidden size | 200 | 200 | 200 |
| Dynamics lr | 1e-2 | 1e-2 | 1e-2 |
| **Our Method** | | | |
| State dim | 3 | 2 | 6 |
| Action dim | 3 | 2 | 2 |
| Batch size | 128 | 128 | 128 |
| Nmodules | 3 | 3 | 3 |
| Discount | 0.99 | 0.99 | 0.99 |
| Optimizer lr | $3 \times 10^{-4}$ | $3 \times 10^{-4}$ | $3 \times 10^{-4}$ |
| Polyak $\tau$ | 0.005 | 0.005 | 0.005 |
| $\psi$ MLP hidden sizes | [256, 256, 256] | [256, 256, 256] | [256, 256, 256] |
| $\phi$ MLP hidden sizes | [256, 256] | [256, 256] | [256, 256] |

the task-specific representations and value predictions across datasets. The $\phi_k$ models are implemented as encoder-decoder networks with two hidden layers of 256 units and the $\psi_k$ models are as critic networks with three hidden layers of 256 units, estimating the Q-values over state-action pairs. We shown hyperparameters in Table 3, which describe hyperparameters that were different across algorithms.

