# OpenReview forum: "Transferring Human Daily Activity Skills to Surgical Robots via Deep Successor Features"
_TMLR — Rejected by TMLR_

### Review · Reviewer_DaJ5 · 2025-11-26

**Summary Of Contributions:**

This paper proposes a framework for transferring motor skills from human Activities of Daily Living (ADL) datasets to surgical robot tasks. The proposed method uses Deep Successor Features (DSF) to factor out task dynamics from reward functions. The authors first train "skill modules" on human data (e.g., spooning, pouring) to learn predictive state representations. Then, in the surgical domain, they learn linear reward weights to adapt these pre-existing modules to surgical tasks (e.g., reaching, tracking) using Generalized Policy Improvement (GPI). The method is evaluated in simulation and on a physical dVRK robot, claiming improved sample efficiency compared to imitation and RL baselines trained from scratch or initialized with naive pretraining.

**Additional Comments:**

Notes on paper clarity:
- The steps described in Section 4's introduction do not map very clearly to figure 4. It may be helpful to the reader to use matching terms or symbols to connect the steps to their corresponding illustration.
- Section 4.1 heading typo: "offlien"
- Eq 10: How do you compute the argmax action? Are the surgical actions represented as continuous or discrete?
- Unclear language "This establishes a scalable and sample-efficient pathway for bridging reusing human ADL task dataset in data-constrained surgical robot task learning." Delete the word "bridging" ?
- Appendix algorithm 1 did not seem to render correctly (steps 4-6)

Questions I had while reading:
- What exactly are the input observations and action space?
- Do you use visual inputs? How are they encoded?
- Does the ADL data include gaze information as part of the action space?

Although I found the answers by the end of the paper or in the appendix, it would be helpful to state these earlier in the body of the paper, to help the reader make better sense of the method description and equations.

**Audience:**

Yes

**Audience Explanation:**

The problem of data scarcity in surgical robotics is significant, and the core idea of using abundant, non-robot data such as ADL to initialize robotic policies is valuable. The application of Successor Features to bridge the embodiment gap between human hands and surgical tools is a novel and interesting angle. Even with the limitations noted above, the proposed method would likely interest researchers in robot learning, transfer learning, and medical robotics.

**Claims And Evidence:**

No

**Claims Explanation:**

Although the proposed method can learn from non-surgical-robot data, the type and amount of data it can use is still far more constrained than what one may expect from the term "activities of daily life". For example it should include the vast amounts of egocentric video data showing humans completing bimanual manipulation tasks in households, offices, kitchens and factories. However, the proposed method at least in its current form requires both robot and object states to be tracked, and cannot operate using robot state and vision alone. This limits the amount and type of non-robot data that can be used for training. Requiring full object state in the policy also seems to severely limit progress towards contact-rich manipulation that would occur in real surgery tasks.

Some specific language in the paper is also too strong in terms of the claims made, e.g.: "Together, these tasks span both manipulation and perception control, providing a realistic and comprehensive benchmark for evaluating the effectiveness and robustness of our method under practical surgical conditions." The experiments are interesting but are still far from a "realistic and comprehensive benchmark ... under practical surgical conditions".

"Our framework enables large-scale offline pretraining on abundant non-surgical datasets" - this claim in the abstract is also not yet supported by the experiments which still rely on small-scaled datasets.

Although the paper is mostly clearly written and experiments are interesting, the claims are currently too strong given the current limitations of the proposed method.

**Requested Changes:**

The authors should narrow down certain claims of the proposed method. In particular, specify exactly what kind of "Activities of Daily Life" datasets can be used by the method.
- The "Daily Interactive Manipulation" (DIM) dataset was used with 1,603 motions across 32 manipulation tasks. Only a subset of 3 was used in the paper experiments. This seems to contradict claim (1) of the paper of "systematically transfers motor skills from large-scale human ADL datasets to surgical robot task learning".
- EgoDex (https://arxiv.org/abs/2505.11709) for example has 829 hours of egocentric video with paired 3D hand and finger tracking data. Can this type of data be used? And if not why not?

The authors should also include more specific details about their model, and the observation and action spaces used, in the main text of the paper rather than in the appendix. Clarify that the policy is not operating from pixels, but rather low-dimensional robot and object states. If possible, indicate how the method could be extended to handle contacts, manipulation and vision-based policies.

---

> ### Author Response · Authors · 2026-01-31
> **Response to Reviewer DaJ5**
>
> We sincerely thank the reviewer for the exceptionally thorough and constructive review. We greatly appreciate the detailed suggestions, which helped us strengthen the empirical components of the paper.
>
> > Although the paper is mostly clearly written and experiments are interesting, the claims are currently too strong given the current limitations of the proposed method. The proposed method at least in its current form requires both robot and object states to be tracked, and cannot operate using robot state and vision alone. This limits the amount and type of non-robot data that can be used for training.
>
> Thank you for the valuable feedback. We agree that several claims in the original manuscript were overly strong given the current experimental scope. We have revised the paper to narrow these statements, including removing “large-scale” and softening the description of the evaluated tasks to avoid framing them as realistic surgical experiments.
>
> Regarding the observation design, while large-scale egocentric video datasets are indeed more accessible and visually rich, the primary goal of this work is to enable rapid and data-efficient transfer to surgical robotic tasks under minimal surgical-domain supervision. Directly leveraging human video data introduces a substantial visual gap, which would significantly increase the need for additional surgical-domain data or online interaction to bridge the mismatch. To isolate and exploit transferable structure at the interaction level, we therefore use object and robot states as observations. **This design allows the framework to focus on motor-level regularities that are more directly shared across human daily activities and surgical manipulation, enabling few-shot or zero-shot adaptation while avoiding confounding perception-related discrepancies.**
>
> > The authors should narrow down certain claims of the proposed method. In particular, specify exactly what kind of "Activities of Daily Life" datasets can be used by the method.
>
> Thank you for the constructive suggestion. We have revised the manuscript to narrow the scope of our claims and explicitly specify the types of Activities of Daily Life datasets applicable to our framework. In particular, we now clarify that in this work, the human daily activity dataset is required to contain motor-level information, specifically the translation and orientation of the tool or human hand.
>
> > The "Daily Interactive Manipulation" (DIM) dataset was used with 1,603 motions across 32 manipulation tasks. Only a subset of 3 was used in the paper experiments. This seems to contradict claim (1) of the paper of "systematically transfers motor skills from large-scale human ADL datasets to surgical robot task learning".
>
> Thank you for raising this concern. While the main experiments initially focused on three representative ADL-derived skill modules for clarity of analysis, the proposed framework supports a larger pool of human activities without manual selection. **To directly address this point, we have added a new ablation study in Section 7.4 that systematically evaluates the effect of using different numbers of skill modules randomly sampled from the full DIM dataset, ranging from 10 to all 32 activities.** The results demonstrate that increasing the number of available modules generally improves learning efficiency, while robust performance can still be achieved with small randomly selected subsets—indicating that the framework effectively exploits transferable structure from large-scale human activity pools without requiring curated task selection. Furthermore, we visualize the temporal activation of all 32 modules during surgical task execution, showing how different human skills are dynamically composed and reweighted depending on the task demands.

---

> > ### Author Response · Authors · 2026-01-31
> > **Response to Reviewer DaJ5**
> >
> > > EgoDex (https://arxiv.org/abs/2505.11709) for example has 829 hours of egocentric video with paired 3D hand and finger tracking data. Can this type of data be used? And if not why not?
> >
> > Thank you for the question. **EgoDex-type datasets with 3D hand tracking can indeed be utilized within our framework, and we explicitly evaluate this in a new ablation study (Section 7.5) comparing EgoDex and DIM datasets.** The results show that tool-centered DIM trajectories consistently yield stronger transfer performance than hand-centered EgoDex data. This performance gap arises from differences in motion representation. EgoDex captures hand movements, whereas DIM records tool-mediated motions that more closely resemble surgical instrument control. While hand and tool trajectories may align in certain tasks, they diverge substantially in many interactions, making tool-centered data more directly transferable for surgical manipulation.
> >
> > > The authors should also include more specific details about their model, and the observation and action spaces used, in the main text of the paper rather than in the appendix. Clarify that the policy is not operating from pixels, but rather low-dimensional robot and object states. If possible, indicate how the method could be extended to handle contacts, manipulation and vision-based policies.
> >
> > Thank you for your comment. Thank you for the helpful suggestion. We have revised the main text to provide more detailed descriptions of the model architecture and explicitly specify the observation and action spaces used in our method.
> >
> > **Our successor feature framework is inherently flexible with respect to the observation space, which can consist of either low-dimensional robot states or high-dimensional visual inputs encoded by deep neural networks.** Prior work has demonstrated the use of visual observations within the successor feature formulation for mobile robot navigation tasks [1], indicating that our modular successor feature framework is naturally compatible with vision-based policies. In this work, we intentionally adopt low-dimensional state representations to enable fast adaptation and minimize online interaction in downstream surgical robot tasks, given the shared motion structure between human daily activities and surgical manipulation.
> >
> > Regarding extension to contact-rich manipulation, **our approach can be readily extended by augmenting the state with contact-relevant variables such as force/torque measurements, gripper states, or constraint-related features.** Successor features can then be learned over this augmented state to capture contact-conditioned transition dynamics. We have discussed these extensions and limitations in the Experiments Discussion and Implications section.
> >
> >
> > [1] Zhang, J., Springenberg, J.T., Boedecker, J. and Burgard, W., 2017, September. Deep reinforcement learning with successor features for navigation across similar environments. In 2017 IEEE/RSJ International Conference on Intelligent Robots and Systems (IROS) (pp. 2371-2378). IEEE.

---

> > > ### Author Response · Authors · 2026-01-31
> > > **Response to Reviewer DaJ5**
> > >
> > > > The steps described in Section 4's introduction do not map very clearly to figure 3. It may be helpful to the reader to use matching terms or symbols to connect the steps to their corresponding illustration.
> > >
> > > Thank you for pointing this out. We have revised Figure 4 to use consistent symbols.
> > >
> > >
> > > > Section 4.1 heading typo: "offlien"
> > >
> > > Thank you for pointing out the typo. We have corrected it in the revised manuscript.
> > >
> > >
> > > > Eq 10: How do you compute the argmax action? Are the surgical actions represented as continuous or discrete?
> > >
> > > Thank you for the comment. The surgical actions are represented as continuous. To compute the argmax action, we sample N candidate actions from the continuous action space and select the action that maximizes the inner product between the successor features and the task-specific reward weights. This procedure has been clarified in the revised manuscript.
> > >
> > >
> > > >  Unclear language "This establishes a scalable and sample-efficient pathway for bridging reusing human ADL task dataset in data-constrained surgical robot task learning." Delete the word "bridging" ?
> > >
> > > Thank you for pointing this out. We have revised the sentence in the manuscript to improve clarity.
> > >
> > >
> > > > Appendix algorithm 1 did not seem to render correctly (steps 4-6)
> > >
> > > Thank you for pointing this out. We have corrected Algorithm 1 in the appendix to ensure that it now renders properly.
> > >
> > >
> > > > What exactly are the input observations and action space? Do you use visual inputs? How are they encoded? Does the ADL data include gaze information as part of the action space? Although I found the answers by the end of the paper or in the appendix, it would be helpful to state these earlier in the body of the paper, to help the reader make better sense of the method description and equations.
> > >
> > > Thank you for the constructive feedback. We have moved and clearly stated the relevant details earlier in the main body of the paper.

---

### Review · Reviewer_76Gq · 2025-11-27

**Summary Of Contributions:**

- A robot learning framework that transfers the robot skills learned on large datasets of daily-life activities to specialized surgical settings.
- Concretely, in the framework, the paper proposes:
	- A state-action representation learning framework to extract the shared features for different tasks;
	- A learned attention mechanism aggregating different skill modules;
	- Integrating these into a learning framework for policy transfer between different MDPs;
	- Theoretical analysis.
- Experiments in both simulation and the real world show the effectiveness of the proposed framework.

**Audience:**

Yes

**Audience Explanation:**

The paper is well-motivated:
- From the application perspective, it addresses the key problem in surgical robots: safety concerns (for RL) and data scarcity (for imitation learning).
- More generally, pre-training on large datasets of general activities/skills and then adapting to more specialized settings is also a widely discussed problem in the robot learning community.
- The inspiration "surgical expertise is not innate but develops gradually" behind the framework also sounds very natural. In fact, I feel among many general -> specialized robot learning/fine-tuning tasks, the problem discussed in this paper is one of the best-motivated ones.

**Broader Impact Concerns:**

None.

**Claims And Evidence:**

Yes

**Claims Explanation:**

I think most key arguments in the paper are well-supported:
- Synthetic experiments with baseline comparisons show that the proposed framework has better performance in multiple aspects.
- Real-world experiments show that the proposed framework can be effectively deployed in the real world.
- Various ablation studies show the effectiveness of different skill modules in pre-training.

**Requested Changes:**

- I feel the paper can be clearer about the assumptions in its problem definition. For example --
	- "Although the two domains differ in embodiment, observation spaces, and task objectives, we hypothesize that they share common low-level motor control structures." What exactly does it mean here with "share common low-level motor control structures"? My understanding is that $(s, a)$ in different MDPs have some relevance, and the work tries to extract that with the representation learning. But what does "low-level" mean here? Also, how does it count into different end-effectors?
	- The representation $f(s, a)$ is not conditioned on the end-effector type? (Also, $s$ is considered to be video frames here? -- so perhaps $f$ implicitly knows what end-effector it is?)
	- All pre-training data are actually human-hand data (instead of tele-operated robot data)? And the end-effector is always the human hand? So that the data for representation pre-training is not assumed to contain different embodiments?

- Experiments:
	- It seems that currently only the proposed framework is tested in the real world? It would be good to also test some baselines (for example, the ablated version by removing some/all skill modules) and compare their results with the proposed method.
	- It seems that the framework is sensitive to the selected skills in the pre-training set (Section 6.3)? But I assume the attention-based aggregation of different skill modules is designed for resolving this problem automatically, instead of hand-selecting what's more relevant to the final application? Could the authors explain more about this?
	- Figure 14: It seems that the single/no-module frameworks also converge to success rate 1.0 in the end -- just taking more time. Is it because the tasks here are too easy for them? Also, in Figure 14 (c), Module 1 is lower than No Module?

---

> ### Author Response · Authors · 2026-01-31
> **Response to Reviewer 76Gq**
>
> We thank the reviewer for the positive and encouraging assessment of our work, as well as for highlighting several limitations that helped us improve the manuscript.
>
> >  What does "low-level" mean here? Also, how does it count into different end-effectors?
>
> Thank you for your comment. In this work, the term “low-level motor control structures” does not imply similarity at the full MDP level (e.g., shared state or action spaces, reward functions, or task objectives). It refers to shared structure at the level of transition dynamics Regarding different end-effectors, we do not assume any direct correspondence between human joint actions and robotic joint commands. Instead, transfer occurs through an end-effector–centric representation, in which the learned features describe predictive motion patterns rather than embodiment-specific actuation. As a result, successor features can leverage common motor regularities even when the underlying effectors and control spaces differ. We have revised the manuscript to explicitly clarify this definition of “low-level motor control structures”.
>
> > The representation is not conditioned on the end-effector type? (Also, is considered to be video frames here? -- so perhaps implicitly knows what end-effector it is?)
>
> Thank you for your comment. In our framework, the learned representation is not explicitly conditioned on the end-effector type, nor is it intended to encode embodiment-specific information such as whether the effector is a human hand or a surgical robotic instrument.
>
> In the experiments presented in this paper, we do not use visual observations as inputs. This is motivated by the large domain gap between human daily activities and robotic surgical tasks, which includes differences in embodiment and visual appearance. Instead, transfer is performed purely at the level of motor information, avoiding high demand on sugircal domain data in downstreme task transfer. The learned representation is trained to capture motor-level transition dynamics, i.e., regularities in how actions induce short-horizon state changes during manipulation.
>
>
> > All pre-training data are actually human-hand data?
>
> Yes. In this work, representation pre-training is performed exclusively using human demonstration data from the DIM dataset, without relying on tele-operated robotic data.
>
>
> > Is the end-effector always the human hand during pre-training?
>
> No. During pretraining, we exclusively use the DIM dataset, which provides tool-centered motion trajectories captured during human daily manipulation. Consequently, the learned skill modules are trained on end-effector–like tool motions rather than human hand motion.
>
>
> > Does this imply that the pre-training data do not include multiple embodiments?
>
> Yes. The DIM dataset does not explicitly include multiple embodiments; it contains only tool-centered motion captured during human daily manipulation. Our framework does not rely on explicit embodiment labels or paired human–robot demonstrations. Instead, embodiment-specific characteristics are implicitly reflected in the motion statistics induced by different kinematic structures and workspaces. This assumption is consistent with prior cross-embodiment transfer studies (e.g., [1]), where successful transfer from human to robot was achieved without requiring multi-embodiment data during pre-training.
>
> [1] Zakka, Kevin, et al. "Xirl: Cross-embodiment inverse reinforcement learning." Conference on Robot Learning. PMLR, 2022.
>
>
> > It seems that currently only the proposed framework is tested in the real world? It would be good to also test some baselines and compare their results with the proposed method.
>
> Thank you for the comment. We agree that evaluating additional baselines in real-world settings would be valuable. However, conducting real-robot experiments is substantially more resource-intensive and constrained by hardware availability, safety considerations, and experiment time. We focus the real-world experiments on validating the deployability and feasibility of the proposed framework on physical robotic systems, rather than on exhaustive baseline comparisons.

---

> > ### Author Response · Authors · 2026-01-31
> > **Response to Reviewer 76Gq**
> >
> > > It seems that the framework is sensitive to the selected skills in the pre-training set (Section 6.3)?
> >
> > Thank you for the insightful question. The attention-based modular aggregation is indeed designed to automatically weight and compose skill modules based on their relevance to the target surgical task, without requiring manual selection. However, its effectiveness depends on the availability of sufficiently informative and transferable skill representations in the pretrained module pool. In Section 7.3, the observed performance drop occurs when the pretraining set includes a dissimilar human activity alongside relevant ones. While the attention mechanism reduces the contribution of less relevant modules, the presence of mismatched dynamics can still introduce representation noise, particularly when the total number of modules is small. When the skill pool contains predominantly relevant activities or is sufficiently large, attention robustly emphasizes useful modules and mitigates the impact of unrelated ones, as demonstrated in later ablations with randomly sampled larger module sets in section 7.4.  We have added this discussion and explanation in the revised manuscript.
> >
> > > Figure 14: It seems that the single/no-module frameworks also converge to success rate 1.0 in the end -- just taking more time. Is it because the tasks here are too easy for them? Also, in Figure 14 (c), Module 1 is lower than No Module?
> >
> > Thank you for these insightful questions. We address the two points separately below.
> >
> > "Why do single/no-module baselines also converge to a success rate of 1.0?"
> >
> > Task difficulty is one contributing factor, but it does not fully explain this behavior. In the single-module setting, each module is pretrained on a human ADL dataset that is related to the target task. Consequently, given enough exploration, both single-module and no-module baselines can asymptotically achieve high success rates.
> >
> > "Why does Module 1 underperform the no-module baseline in Fig. 14(c)?"
> >
> > Figure 13(c) corresponds to the ECM dynamic tracking task, which differs fundamentally from the static localization tasks shown in Fig. 13(a,b). Dynamic tracking requires continuous temporal coordination and sustained motion control, making it more sensitive to the diversity and coverage of the source skill data.  The successor features learned from one single dataset provide limited coverage of the state–action space required for accurate and robust tracking behavior. In contrast, the no-module baseline is trained on a mixture of all ADL datasets. This dataset aggregation increases trajectory diversity and partially compensates for the missing or underrepresented behaviors present in Module 1’s single-source dataset.
> >
> > However, simply mixing datasets and training a single monolithic model is not a principled or scalable solution. While dataset mixing may suffice for relatively simple tasks, it cannot effectively resolve conflicting or complementary skill structures across diverse source domains. In contrast, our proposed skill modularization framework explicitly preserves individual skill representations and adaptively modulates their contributions, enabling the policy to compensate for deficiencies in any single dataset.  We have discussed these experimental performance in the revised manuscript.

---

### Review · Reviewer_VwZm · 2026-01-11

**Summary Of Contributions:**

The authors introduce a pipeline to pretrain and transfer motor primitives from daily activities data to a surgical robotics application through the use of successor features. This alleviates the problem of limited demonstrations or data for critical applications such as surgeries through pretraining and transfer learning. The authors demonstrate that their method outperforms two baselines on simulated and real-world tasks in surgical robotics, and provide theoretical results on performance guarantees of the transfer learning approach. They also carry out an ablation study where the claim is that the specific modular and compositional implementation of the skills or motor primitives improves performance quicker and in a more stable manner over the non-modular and non-compositional alternatives. The authors also attempt to provide some interpretability on the activation dynamics of the various motor primitives and show that human daily activities contain a diverse set of motions that would transfer to surgical applications.

**Strengths:**
1. The application considered is useful and the method of pretraining + transfer seems principled.
2. The authors have provided theoretical results on performance guarantees of their method.
3. The method seems to perform well in real-world experiments as well.

**Weaknesses:**
1. The work seems to be a very simple extension with not much novelty over Hu, Tavakoli & Jin (2025). Several figures seem to be shared or mildly edited versions of those in that paper. There are a few more tasks considered in this version, and the baselines considered are different. However, I have serious concerns on the originality and whether the work included here meets the originality bar for TMLR: simple extensions of prior published work are not accepted at TMLR to my knowledge. Could the authors clarify the differences and the exact novelty over the prior work?
2. The baselines considered here seem very simplistic. There are several other works which have explored the use of motor primitives or skills for robotics applications and in reinforcement learning in general, e.g., Rana et al. (2023) or Nasiriany, Liu & Zhu (2022).
3. The figures and presentation require a lot of polishing. See requested changes for some details. Several figures are either not clear, or it is not well-explained what is being shown in the figure, or it is not clear what purpose the figure serves. Examples include Figure 13 and 16 which lack clarity; Figure 2 should be a table; Figure 1 serves no real purpose and is not very clear; etc.

**References:**

Hu, Yi, Mahdi Tavakoli, and Jun Jin. "Pretraining Using Comparable Human Activities of Daily Living Dataset in Robotic Surgical Task Learning." IEEE Transactions on Medical Robotics and Bionics (2025).

Rana, Krishan, et al. "Residual skill policies: Learning an adaptable skill-based action space for reinforcement learning for robotics." Conference on Robot Learning. PMLR, 2023.

Nasiriany, Soroush, Huihan Liu, and Yuke Zhu. "Augmenting reinforcement learning with behavior primitives for diverse manipulation tasks." 2022 International Conference on Robotics and Automation (ICRA). IEEE, 2022.

**Audience:**

Yes

**Audience Explanation:**

The paper could be of interest to TMLR's audience as it discusses the application of existing technique, i.e., pretraining and successor features, to a useful real-world problem. However, as stated previously, I have concerns about the originality of the submission and whether it conforms to TMLR guidelines.

**Broader Impact Concerns:**

I have no concerns apart from those already addressed in the Conclusions and Limitations section.

**Claims And Evidence:**

No

**Claims Explanation:**

I am not convinced that the claims made in the submission are well supported for the following reasons:
* The novelty is questionable given highly overlapping prior work, and thus whether the submission even follows the guidelines for TMLR is unclear to me.
* The baseline methods compared are very simplistic and there are no alternative implementations of pretrained or primitive-based methods such as those cited above in my review, apart from others.
* The theoretical results are very brief, no intuition or clear explanation is provided to support them and state their practical utility. Given the number of terms in the bounds, I would have expected a study or some plots to help interpret the resulting equations.
* The interpretability studies are not clear enough to actually serve as interpretations.

**Requested Changes:**

* Could the authors clarify what exactly is the novel technical contribution over Hu, Tavakoli & Jin (2025)?
* The figures and presentation need to be spruced up. In particular, it's not clear what value Figure 1 adds, Figure 2 should be a table, the text in other figures such as the experimental results needs to be larger, Figures 9 (b, c) are not clear, Figure 13 is not easily interpretable and is unclear, similarly Figure 16 does not elucidate what the trajectories are or provide any interpretability, etc. Significant work is required to improve the figures and explanation of results.
* In Figure 10d, a "consistent" downwards trend is claimed but this is not evident or at least not consistently so. Could the authors comment on this?
* Several other works have explored skills or primitives in the context of robotics and reinforcement learning, however the baselines compared here are quite simplistic. Could the authors add additional baselines, including those incorporating skills in different ways, e.g., Rana et al. (2023) or others?

**References:**

Hu, Yi, Mahdi Tavakoli, and Jun Jin. "Pretraining Using Comparable Human Activities of Daily Living Dataset in Robotic Surgical Task Learning." IEEE Transactions on Medical Robotics and Bionics (2025).

Rana, Krishan, et al. "Residual skill policies: Learning an adaptable skill-based action space for reinforcement learning for robotics." Conference on Robot Learning. PMLR, 2023.

Nasiriany, Soroush, Huihan Liu, and Yuke Zhu. "Augmenting reinforcement learning with behavior primitives for diverse manipulation tasks." 2022 International Conference on Robotics and Automation (ICRA). IEEE, 2022.

---

> ### Author Response · Authors · 2026-01-31
> **Response to Reviewer VwZm**
>
> We sincerely thank the reviewer for the insightful questions and suggestions. We have addressed all concerns raised by the reviewer and revised the manuscript in line with comments.
>
> > Several other works have explored skills or primitives in the context of robotics and reinforcement learning, however the baselines compared here are quite simplistic. Could the authors add additional baselines, including those incorporating skills in different ways, e.g., Rana et al. (2023) or others?
>
> Thank you for the thoughtful suggestion. We designed our baselines to explicitly capture a spectrum of transfer settings in order to isolate and evaluate the practical benefits of leveraging human ADL data for surgical robotic learning. Specifically, (1) COMBO represents a pure transfer scenario that utilizes prior data without further task-specific adaptation; (2) MBPO serves as a non-transfer baseline trained entirely from scratch on surgical tasks; and (3) MBPO+ADL corresponds to partial transfer, where models are initialized using ADL data and subsequently adapted to surgical tasks. Many pretrained or primitive-based approaches, including those cited in the comment, follow a similar paradigm to MBPO+ADL, namely pretraining on a source dataset followed by fine-tuning on downstream tasks. **Our baseline setting is intentionally structured to compare pure imitation-style transfer, training learning from scratch, and ADL-pretrained adaptation, providing a controlled evaluation of human-to-surgical skill transfer**.
>
>
> > The theoretical results are very brief, no intuition or clear explanation is provided to support them and state their practical utility. Given the number of terms in the bounds, I would have expected a study or some plots to help interpret the resulting equations.
>
> Thank you for the comment. The original successor feature framework establishes performance guarantees for policy transfer by exploiting shared structure across tasks [1]. In this work, we extend this principle to a modular successor feature formulation to model transferable dynamics between human ADL tasks and surgical robotic tasks. **The intuition behind the theoretical results is to formally characterize when and why ADL-derived representations enable reliable policy adaptation.**  To improve interpretability, we have expanded the explanation of these results in the revised manuscript
>
> [1] Barreto, André, et al. "Successor features for transfer in reinforcement learning." Advances in neural information processing systems 30 (2017).

---

> > ### Author Response · Authors · 2026-01-31
> > **Response to Reviewer VwZm**
> >
> > > The interpretability studies are not clear enough to actually serve as interpretations.
> >
> > Thank you for the comment. Our experiments and ablation studies are explicitly designed to analyze how human ADL–derived skill modules contribute to surgical task performance within the successor feature framework.
> >
> > Concretely, we visualize the temporal activation of individual skill modules during task execution, revealing how human activity representations are engaged across different surgical tasks. We further conduct systematic ablations on module composition, ADL task selection, dataset source, and the number of skill modules, demonstrating how each factor influences transfer efficiency, convergence behavior, and final performance. In addition, we provide theoretical analysis characterizing how discrepancies between source ADL tasks and target surgical tasks affect policy transfer under the successor feature formulation, offering formal insight into the conditions under which transfer remains effective.
> >
> > > The figures and presentation need to be spruced up. In particular, it's not clear what value Figure 1 adds, Figure 2 should be a table, the text in other figures such as the experimental results needs to be larger, Figures 9 (b, c) are not clear, Figure 13 is not easily interpretable and is unclear, similarly Figure 16 does not elucidate what the trajectories are or provide any interpretability, etc. Significant work is required to improve the figures and explanation of results.
> >
> > Thank you for the comment. Figure 1 is intended to illustrate that surgical skills are progressively developed from human daily tool-use behaviors rather than being innate. We have revised Figure 2 into a table for clearer presentation. Figures 8(b,c) present raw endoscopic camera views and are included to provide representative visual observations rather than processed results. We have revised Figures 13 and 16 to improve clarity and detailed the explaination of results.
> >
> > > In Figure 10d, a "consistent" downwards trend is claimed but this is not evident or at least not consistently so. Could the authors comment on this?
> >
> > Thank you for pointing this out. Figure 9(d) reports the number of steps required to reach different target gauze positions under varying initial robot states, where each data point corresponds to a distinct initial–target configuration. Since the ADL dataset does not uniformly cover the full state–action space relevant to the surgical task, the transferred representations may generalize better to some regions of the state space than others. As a result, early performance across different targets can exhibit noticeable variability rather than a strictly monotonic decrease. As the testing number of the target gauze increases, we expect these local fluctuations to average out, revealing a clearer overall downward trend that reflects improving task efficiency during adaptation. To better reflect this behavior, we have revised the wording in the manuscript from “consistent downward trend” and added further explanation to clarify the source of the observed fluctuations.

---

> ### Author Response · Authors · 2026-02-01
> **Response to Reviewer VwZm**
>
> > The work seems to be a very simple extension with not much novelty over Hu, Tavakoli & Jin (2025).  Could the authors clarify what exactly is the novel technical contribution over Hu, Tavakoli \& Jin (2025)?
>
> Thank you for raising this important point. While both works explore the high-level idea of transferring human daily activity skills to robotic surgical task learning, the technical contributions and scope of our work differ fundamentally from Hu, Tavakoli & Jin (2025).
>
> The prior work, which is based on **successor features combined with probabilistic movement primitives**, depends on a **manually curated set of “relevant” human activities** and selects a single most relevant activity from this candidate pool as the transferable policy. This design limits scalability and robustness, particularly when task relevance is ambiguous or shared across multiple skills. It further relies on **hand-designed feature representations $\phi$** for human activities, thereby constraining the expressiveness and adaptability of the learned representations across diverse manipulation behaviors.
>
> In contrast, our work introduces a **modular successor feature framework** with learned representations and structured multi-skill composition. We jointly **learn deep feature representations $\phi$** from human ADL data together with their corresponding **successor features $\psi$** using both feature learning losses and successor feature Bellman objectives. Each human activity is encoded as a skill module. During downstream surgical task learning, instead of assuming the existence of a single transferable skill, our framework automatically assigns task-conditioned weights across multiple pretrained modules, enabling **dynamic multi-skill composition** tailored to each surgical objective.
>
> Beyond this framework structure shift, the two works differ substantially in several key aspects:
>
> - **Theoretical analysis**: We provide formal performance bounds and transfer guarantees for modular successor feature adaptation, which were different from the prior work.
>
> - **Experimental design and baselines**: Experimental design and baselines: We conduct a comprehensive simulation evaluation across three distinct surgical tasks, including one PSM manipulation task and two ECM perception–control tasks, and introduce structured baselines representing pure transfer, learning from scratch, and ADL-pretrained adaptation. The prior work which considered only a single PSM task with limited baseline comparison.
>
> - **Real-world validation**: We validate the proposed multi-skill transfer framework across three real-world tasks, extending beyond the scope of the prior study.
>
> - **Comprehensive ablation studies**: We conduct five systematic ablations across ***module design, ADL task subsets, dataset sources, number of modules, and temporal activation behavior, providing actionable insights into robustness and generalization*** when transfering from human to surgical robot.
>
> Importantly, in Section 7.2 we explicitly reproduce the single-skill transfer setting as a special case of our framework by restricting the model to a single skill module. **The results in Fig.13 show that the proposed modular design consistently outperforms this single-skill baseline, achieving at least 55.2%, 40.8%, and 41.4% improvements in sample efficiency** on the PSM reaching task, ECM static localization task, and ECM dynamic tracking task, respectively.
>
> Overall, while the motivation of leveraging human daily skills is shared, **our work advances the field from manual single-skill reuse to principled multi-skill composition. In addition to this technical advancement, our work provides a systematic and reproducible study of robustness and generalization in human daily activities to robotic surgical skill transfer.** We have explicitly clarified these distinctions and discussed the comparison in the revised manuscript, particularly in the Introduction, Related Work, and Ablation Study sections.

---

> > ### Comment · Reviewer_VwZm · 2026-02-23
> >
> > Thank you for the response. In particular, thank you for the added context in certain parts of the paper, the additional experiments with more skills and random subsets, and the EgoDex dataset experiment.
> >
> > **Comments:**
> > * Your baselines cover (1) training from scratch, (2) imitation-style transfer, and (3) adapting ADL-based models to the surgical task. I agree that they cover a set of different transfer scenarios, however, they do not represent the set of existing skill-based methods that could be tested from scratch or be transferred across settings. While skill-based methods may be quite similar to (3), the existing baselines are not sufficient given the breadth of skill-based methods that exist in the literature (e.g., Nasiriany et al., 2022; Rana et al., 2023; Pertsch et al., 2020 – SPiRL). The proposed method thus needs to be compared to existing approaches, e.g., skill-based methods that don't rely on learning successor features (such as SPiRL, which learns a latent skill embedding space with a learned skill prior from offline data – directly comparable to the ADL pretraining setting here), specifically to test if the proposed approach is better than those alternative proposals.
> > * Thank you for the additional explanation on the theoretical results.
> > * The interpretability results and response are unfortunately still not clear. The module activation plots (Figure 12) show that module-wise contributions fluctuate over time, but fluctuation alone does not equal interpretability. There is no grounded explanation of why a particular module activates at a particular time step, e.g., whether it corresponds to an identifiable motor phase. Without a mapping from activation patterns to semantically meaningful task phases, the claim of interpretability is not substantiated. There is also a lot of variance across instances of the same task, could you plot averages or clarify this?
> > * Related to Figure 1, simply honing daily skills may not necessarily translate to surgical skills, the latter are more precise. This point relates further to another limitation which is that the tasks considered are among the most simple surgical-like tasks. Furthermore, pretraining of some kind generally benefits downstream transfer. To show that human activity skills are specifically beneficial, the control would be completely different pretraining, e.g., pretraining on random motor trajectories, on locomotion data, or on non-manipulation tasks. The skill subset experiments (Section 7.3) actually show that only certain skills chosen carefully are most useful, and the EgoDex ablation (Section 7.5) shows that hand-centered data transfers poorly compared to tool-centered data. These results undercut the narrative that ADL data is generically useful and instead suggest that specific kinematic similarity is doing the heavy lifting. This distinction matters for the paper's central claim.
> > * Would it be possible to see experimental evidence for the claim associated with "consistent downwards trend" of the number of steps required? Unfortunately without evidence, it would seem to me that the claim remains speculative.
> > * Thank you for clarifying this, I appreciate that the paper now includes a discussion given the strong relationship with Hu, Tavakoli & Jin (2025). In light of this, claims such as this proposal being the "first" to transfer skills from ADL tasks to surgical robotics tasks should be reformulated. I would also strongly encourage the authors to reframe claims of surgical relevance, in line with Reviewer DaJ5's suggestions, given the much simpler nature of tasks considered here.
> >
> > **Additional comments:**
> > * Several figures remain difficult to read. For example, most figures, especially Figures 12 and 17, have tiny axis labels and the latter lacks a clear colour-bar legend. Figure 6's y-axes are inconsistently scaled across the three subplots, which looks awkward. Figure 15 remains hard to understand – there are several trajectories but it is hard to visualise the dynamics as it is too cluttered and there is no temporal colouring. Despite the revisions mentioned, the figures still need further work for a venue like TMLR.
> > * The manuscript still contains some grammatical errors and typos, e.g., "embiodment", "actvitites", "teh", "reresentation" (these were on Page 10). I request the authors to proofread the text and fix these issues.
> >
> > **References:**
> >
> > Nasiriany, Soroush, Huihan Liu, and Yuke Zhu. "Augmenting reinforcement learning with behavior primitives for diverse manipulation tasks." 2022 International Conference on Robotics and Automation (ICRA). IEEE, 2022.
> >
> > Rana, Krishan, et al. "Residual skill policies: Learning an adaptable skill-based action space for reinforcement learning for robotics." Conference on Robot Learning. PMLR, 2023.
> >
> > Pertsch, Karl, Youngwoon Lee, and Joseph Lim. "Accelerating reinforcement learning with learned skill priors." Conference on robot learning. PMLR, 2021.

---

### Author Response · Authors · 2026-01-31
**We have updated our PDF and responded to reviewers**

Dear Esteemed Reviewers and Action Editor,

We sincerely thank you for your thoughtful feedback and constructive comments. We have replaced our previous submission with a thoroughly revised manuscript, in which all modifications are highlighted for clarity. In addition, we provide a separate revision summary to facilitate a quick overview of the changes.

In the revised paper, we have carefully addressed the reviewers’ concerns by refining overly broad claims, clarifying methodological details, and adding new experimental analyses. The major updates are summarized as follows:

- **Abstract**: Narrowed and refined several claims of the proposed method.

- **Introduction**: Refined key claims and added discussion of related work.

- **Related Work**: Added detailed comparisons with prior studies and revised the original figure into a table for clearer presentation.

- **Background and Problem Setup**: Clarified the definition of the term *“low-level.”*

- **Method**: Improved terminological clarity; explicitly specified the types of ADL datasets applicable to our framework; and corrected Figure 3.

- **Theoretical Analysis**: Expanded the explanation and intuition behind the theoretical results.

- **Experiments**: Clearly described the model architecture and observation/action spaces, and provided more detailed interpretation of experimental outcomes.

- **Ablation Studies**:
  - Added experiments using a larger number of human skill modules
  - Added experiments using the EgoDex dataset
  - Discussed potential extensions to contact-rich manipulation tasks
  - Clarified that the pretraining data consist solely of human data
  - Expanded discussion of Figures 13 and 14

- **Appendix**: Corrected Algorithm 1


Sincerely,
The Authors

---

### Decision · Action_Editor_E2QA · 2026-02-25

**Recommendation:** Reject

**Additional Comments:**

The reviewers and the action editor agree that the premise of the paper is highly compelling. However, significant development is required to meet TMLR's criteria for evidence and support of claims. If the authors choose to undertake a major revision and resubmit in the future, the following changes are strongly recommended:

More convincing evaluation: Evaluate the method on tasks that actually involve contact-rich manipulation.

Stronger baselines: Include comparisons against state-of-the-art skill-based RL methods (e.g., SPiRL, Nasiriany et al. 2022, Rana et al. 2023).

Clarify data scaling: If claiming the use of "abundant" or "large-scale" datasets, demonstrate the framework working with actual large-scale offline datasets (e.g., incorporating visual inputs or larger egocentric datasets that bridge the visual/state gap without relying purely on low-dimensional tracked states).

**Audience:**

Yes

**Audience Explanation:**

The core motivation of the paper is very strong. Addressing data scarcity and safety constraints in surgical robotics by leveraging offline, non-surgical human data (like ADLs) is highly relevant. Researchers in robot learning, transfer learning, and medical robotics would undoubtedly be interested in exploring principled ways to bridge the embodiment and domain gaps between human daily activities and robotic surgery.

**Claims And Evidence:**

No

**Claims Explanation:**

While the paper addresses a highly motivated and interesting problem: transferring skills from human Activities of Daily Living (ADL) to surgical robots. However, the claims are not adequately supported as pointed out by multiple reviewers (DaJ5, VwZm, Waw8).

Key issues are:

Misalignment between claims and empirical scope: The paper claims to enable "offline pretraining on abundant non-surgical datasets." However, the experiments rely on a single, relatively small dataset (DIM, containing only 1,603 motion trajectories) and utilize low-dimensional robot and object states rather than vision. Furthermore, the evaluated tasks (reaching, static localization, etc) lack the contact-rich manipulation critical for actual surgical procedures.

Insufficient Baselines: Reviewers pointed out that the baselines evaluated (learning from scratch, simple imitation-style transfer) are overly simplistic. The paper fails to compare the proposed method against established skill-based reinforcement learning and transfer methods (e.g., SPiRL, Residual Skill Policies). This makes it difficult to assess the advantage of the proposed approach w.r.t prior methods.

**Resubmission Of Major Revision:**

The authors may consider submitting a major revision at a later time.